# The impact of uncertainty on optimal emission policies

Nicola Botta[1], Patrik Jansson[2], and Cezar Ionescu[3]

[1]Potsdam Institute for Climate Impact Research, Telegraphenberg A31, 14473 Potsdam, Germany
[2]Chalmers University of Technology, Computer Science and Engineering, SE-412 96 Göteborg, Sweden
[3]University of Oxford, Dept. for Continuing Education, University of Oxford, Ewert House, Ewert Place, OX2 7DD

*Correspondence to:* botta@pik-potsdam.de

**Abstract.** We apply a computational framework for specifying and solving sequential decision problems to study the impact of three kinds of uncertainties on optimal emission policies in a stylized sequential emission problem. We find that uncertainties about the implementability of decisions on emission reductions (or increases) have a greater impact on optimal policies than uncertainties about the availability of effective emission reduction technologies and uncertainties about the implications of trespassing critical cumulated emission thresholds. The results show that uncertainties about the implementability of decisions on emission reductions (or increases) call for more precautionary policies. In other words, delaying emission reductions to the point in time when effective technologies will become available is sub-optimal when these uncertainties are accounted for rigorously. By contrast, uncertainties about the implications of exceeding critical cumulated emission thresholds tend to make early emission reductions less rewarding.

## 1 Introduction

### 1.1 About this work

In this article we apply the framework for specifying and solving sequential decision problems (SDPs) presented in Botta et al. (2017b) to understand the impact of uncertainty on optimal greenhouse gas (GHG) emission policies. Specifically, we study the impact of

1. Uncertainty about the implementability of decisions on GHG emission reductions,

2. uncertainty about the availability of efficient technologies for reducing GHG emissions,

3. uncertainty about the implications of exceeding a critical threshold of cumulated GHG emissions.

The work is also an application of the computational theory of policy advice and avoidability proposed in Botta et al. (2017a). The theory supports a seamless approach towards accounting for different kinds of uncertainties and makes it possible to rigorously assess the logical consequences, among others, the risks, entailed by the implementation of optimal policies. We explain what policies are and what it means for a policy sequence to be optimal in section 2.3.

## 1.2 Sequential decision problems and climate change

In many decision problems in the context of climate change, decisions have to be taken *sequentially*: emission rights are issued year after year, emission reduction plans and measures are iteratively revised and updated at certain (perhaps irregular) points in time, etc.

In its fourth Assessment Report on Climate Change (2007), the Intergovernmental Panel on Climate Change (IPCC) has pointed out that responding to climate change involves "an iterative risk management process that includes both mitigation and adaptation, taking into account actual and avoided climate change damages, co-benefits, sustainability, equity and attitudes to risk."

The paradigmatic example of iterative SDPs in the context of climate change is that of controlling GHG emissions. In GHG emission control problems, a decision maker or a finite number of decision makers (countries) have to select an emission level or, equivalently, a level of emission abatement (reduction) with respect to some reference emissions. The idea is that the selected abatement level is then implemented, perhaps with some deviations, over a certain period of time. After that period another decision is taken for the next time period.

Implementing abatements implies both costs and benefits. These are typically affected by different kinds of uncertainties but the idea is that, for a specific decision maker, a significant part of the benefits come from avoided damages from climate change. Avoided damages essentially depend on the overall abatements: higher global abatements lead to less damages and thus higher benefits. In contrast, costs are very much dependent on the abatement level implemented by the specific decision maker. Here, higher emission reductions cost more than moderate emission reductions.

It turns out that, when considering a single decision step and for fairly general and realistic assumptions on how costs and benefits depend on abatement levels, the highest global benefits are obtained if all decision makers reduce emissions by certain "optimal" amounts (Finus et al., 2003; Helm, 2003; Heitzig et al., 2011).

In this situation, however, many (if not all) decision makers typically face a free-ride option: they could do even better if they themselves would not implement any emission reduction (or, perhaps, if they would implement less reductions) but all the others would still comply with their quotas. It goes without saying that if all players fail to comply with their optimal emission reduction quotas, the overall outcome will be unsatisfactory for all or most players.

This situation is often referred to as an instance of the "Tragedy of the Commons" Hardin (1968) and has motivated a large body of research, among others, on coalition formation and on the design of mechanisms to deter free-riding. These studies are naturally informed by game-theoretical approaches and focus on the non-parametric nature of decision making. The sequentiality of the underlying decision process and the temporal dimension of decision making are traded for analytic tractability. For a survey, see Heitzig et al. (2011).

Another avenue of research focuses on the investigation of optimal global emission paths or, as we shall see in section 2.3, of optimal sequences of global emission policies. Here, the core question is how uncertain future developments, typically, the introduction of new technologies or the crossing of climate stability thresholds, shall inform current decisions. In a nutshell,

the problems here are *when* global emissions should be reduced and by *how much* given the uncertainties that affect both our understanding of the earth system and the socio-economic consequences of implementing emission reductions.

In these kinds of studies, the presence of multiple decision makers with possibly conflicting interests and the question of *how* emission reductions can actually be implemented is neglected. This makes it possible to apply control theoretical approaches and to fully account for the temporal dimension of sequential emission games. This is also the approach followed in this work. To the best of our knowledge, no theory is currently available for tackling the problem of computing optimal emission policies for individual countries as a (mixed sequential and simultaneous) coordination game with a finite number of decision makers, over a finite (but not necessarily known) number of decision steps and under different sources of uncertainty. For a survey of SDPs under uncertainty in climate change see Parson and Karwat (2011), Sonja (2006) and references therein.

## 1.3 Stylized sequential emission problems

One can try to understand the impact of uncertainties on optimal emission policies for a specific, real (or, more likely, realistic) emission problem. This requires, among others, specifying an integrated climate-economy assessment model or, as done in Webster (2008), some tabulated version of the model underlying the problem. The approach supports drawing conclusions which are specific for the problem under investigation and is what is typically done in applied policy advice. On the other hand, studying a specific, realistic problem makes it difficult to draw general conclusions and is well beyond the scope of this work.

An alternative approach towards understanding the impacts of uncertainties on optimal policies is to study a "stylized" emission problem. A stylized emission problem does not attempt at being realistic. Instead, it tries to capture the essential features of a whole class of problems and supports general instead of specific conclusions. This is the approach followed in this paper.

## 1.4 Notation

In section 5 we apply the theory for specifying and solving SDPs from Botta et al. (2017b, a) to the stylized emission problem from section 4. The theory is based on the notion of *monadic* dynamical systems originally introduced in Ionescu (2009). In this context, monads allows one to treat deterministic, non-deterministic, stochastic, fuzzy, etc. uncertainty with a seamless approach: the differences are captured by a single problem parameter and all computations are generic with respect to this parameter. In a nutshell, the theory is a dependently typed formalization of dynamic programming (Bellman, 1957). The formalization language is Idris, see Brady (2013). For a discussion on why functional, dependently typed languages are the first choice for implementing such formalizations, see Botta et al. (2017a).

Because the theory is dependently typed, some familiarity with a functional, dependently typed notation is mandatory to apply it to a specific decision problem. In this paper, we do not assume that our readership is familiar with dependent types and functional languages. Thus, in sections 2 to 6 , we have restricted the formalism to the barely minimum. A simplified summary of the Botta et al. (2017a) theory is provided in appendix A.

Still, a number of formulas appear in sections 2 to 6. In the rest of this section we introduce the notation used in these formulas. This is a blend of standard mathematical notation and of standard (Haskell, Idris, Agda, etc.) functional programming notation.

Thus, for instance, in section 2, we write $Technology = \{Available, Unavailable\}$ to posit that $Technology$ is a set that consists of two elements: $Avaialable$ and $Unavailable$. This is plain set comprehension as in $Bool = \{False, True\}$, $A = \{7, 4, 2\}$ or $Even = \{2 * n \mid n \in \mathbb{N}\}$.

Further, in section 2, we write $State : (t : \mathbb{N}) \rightarrow Type$ to posit that "$State\ t$ denotes the set of states the decision maker can observe at the $t$-th decision step". This is now standard Idris notation. Idris (and Haskell, Agda) follows the usual meaning of parentheses in mathematics: to enclose a sub-expression to resolve operator precedence. The special notation $f(a)$ for the value of a function $f : A \rightarrow B$ at $a \in A$ (very much used in physics and engineering) uses parentheses in a non-standard way.

Another possible source of confusion is the signature (type) of the function $State$. Its domain are values of type $\mathbb{N}$ that is, natural numbers. But its co-domain are values of type $Type$! Thus, for instance a legal definition of $State$ could be

$$State\ t = Bool$$

which posits that $State$ is the constant function that returns the type $Bool$ for every $t$. Being able to implement functions that return types is a key feature of dependently typed languages. Among others, it allows one to encode first-order logic propositions as types. Thus, for instance

$$BoundedBy : \mathbb{N} \rightarrow List\ \mathbb{N} \rightarrow Type$$
$$BoundedBy\ n\ ms = All\ (\lambda m \Rightarrow m \sqsubseteq n)\ ms$$

is a legal function definition and a value of type $BoundedBy\ 5\ xs$ is equivalent to a logical proof that all elements of $xs$ are smaller or equal to 5. Perhaps unexpectedly, the type of $m \sqsubseteq n$ on the right hand side of the definition of $BoundedBy$ is $Type$, not $Bool$. This also the type of $All\ (\lambda m \Rightarrow m \sqsubseteq n)\ ms$ as can be seen from the declaration of $BoundedBy$. Thus, $m \sqsubseteq n$ and $All\ (\lambda m \Rightarrow m \sqsubseteq n)\ ms$ encode logical propositions as types: they are propositional types. Here, $(\sqsubseteq)$ and $All$ are standard data types defined in the Idris libraries. Being able to encode logical propositions as types is crucial for formulating program specifications, that is, properties that a program must satisfy to be correct. Thus, for example, a program

$$sqrt : Double \rightarrow Double$$

that is meant to compute the square root of a double precision floating point number might be required to fulfill

$$sqrtSpec : (x : Double) \rightarrow 0 \sqsubseteq x \rightarrow (sqrt\ x) * (sqrt\ x) = x$$

The specification $sqrtSpec$ is logically equivalent to the proposition "for all $x$ of type $Double$, if $x$ is non-negative then the square of $sqrt\ x$ is equal to $x$"[1]. As above, $0 \sqsubseteq x$ and $(sqrt\ x) * (sqrt\ x) = x$ are values of type $Type$, in contrast to $0 \leqslant x$ and $(sqrt\ x) * (sqrt\ x) == x$ which are Boolean values. They encode properties that depend on a specific value $x$ that is, they are dependently typed types.

---

[1] A more realistic specification would require that the square of $sqrt\ x$ is equal to $x$ up to round-off errors but we do not insist on these details here.

The types of *sqrt* and *sqrtSpec* formulate a well defined, unambiguous task for the programmer. This is solved by providing implementations of *sqrt* and *sqrtSpec* that are syntactically correct and total. In this case, the implementation of *sqrt* is said to be verified or, equivalently, machine checked. Notice that totality plays a crucial role in this context. Only total implementations of *sqrt* and *sqrtSpec* are logically equivalent to the proposition "for all $x$ of type *Double*, if $x$ is non-negative then the square of *sqrt* $x$ is equal to $x$".

Because *sqrtSpec* represents a property of *sqrt*, implementations of *sqrtSpec* will depend on a specific implementation of *sqrt*: one typically starts by implementing *sqrt* and then tries to prove that that implementation is correct that is, to implement *sqrtSpec*. Implementations of *sqrtSpec* are typically derived from pencil-and-paper proofs (that *sqrt* fulfills the property encoded in the type of *sqrtSpec*) but dependently typed languages provides a lot of support to programmers: the Idris system can "fill in" parts of the implementation of *sqrtSpec* automatically.

Dependent types are also the key for expressing (perhaps non-implementable) modeling assumptions or conjectures and for formalizing domain-specific notions precisely. In sections 2 to 6 we will not make explicit usage of propositional types. But propositional types are at the core of the theory presented in Botta et al. (2017b, a) and are extensively used there and in appendix A.

Another perhaps unfamiliar aspect of functional notations is currying. In mathematics, a function of $n > 1$ arguments is often implicitly converted to a function that takes as a single argument one $n-$tuple. In Idris we instead use nested function application. Thus, if $g$ has type $X \rightarrow (Y \rightarrow Z)$ or simply $X \rightarrow Y \rightarrow Z$ we write $(g\ x)\ y$ (or simply $g\ x\ y$ because function application is left-associative) to denote the value (of type $Z$) of $g\ x$ (a function of type $Y \rightarrow Z$) at $y : Y^2$.

Notice that even though we do not use propositional types in 2 to 6, most functions there are dependently typed. Thus, for instance, in the signature of *Control* at page 5, the type of the second argument, *State t*, depends on the value of the first argument, $t$. We say that *Control* is dependently typed (Norell, 2007; Brady, 2013, 2017).

Finally, the Botta et al. (2017b, a) theory applied in this paper is available in the `SequentialDecisionProblems` component of Botta (2016–2017). This is a git repository and it is publicly available.

## 1.5  Outline

In the next section we introduce sequential emission problems and explain what it means for sequences of emission policies to be optimal. We discuss the most important differences between deterministic (certain) problems and emission problems under uncertainty. In section 3 we discuss some important traits of decision making under uncertainty. The discussion is meant to prepare the specification of the stylized emission problem presented in section 4. In section 5 we study the impact of the uncertainties (1)–(3) on optimal policy sequences for our stylized problem. We draw preliminary conclusions and outline future work in section 6.

---

[2]The idea that functions of more than one variable can always be written as functions of just one variable (that return functions as result) was originally proposed in Schönfinkel (1924) and popularized by Haskell B. Curry (1958). The operation is since the referred to as *currying*. Its inverse is called *uncurrying*.

## 2    Sequential emission problems

As anticipated in the introduction, in this work we study the impact of uncertainties on optimal emission polices from a control theoretical (as opposed to a game theoretical) perspective. Thus, the focus is on a single decision maker and on how uncertainties affect the questions of *when* global emissions shall be reduced and by *how much* as opposed to the question of *how* emission reductions can actually be implemented in a situation of mutual competition.

### 2.1    Sequential emission processes

If we focus the attention on a single decision maker and on global emissions, sequential emission problems can be described quite straightforwardly. At the core of any such problems one has a sequential emission *process* (SEP). Informally, a sequential emission process can be described in terms of three notions.

The first notion is that of a *state*. A state represents the information available to the decision maker at a given decision step. Typically, the state of a decision process consists of a number of aggregated measures. For instance, economic growth measures, GHG concentration measures, current emission level, etc.

Often, the information available to the decision maker is imperfect. For instance, for a given measure, the decision maker might only be able to know a probability distribution instead of a precise value. Another possibility is that the decision maker only knows that, e.g., a GDP measure lies within certain bounds.

In the stylized sequential emission problem discussed in section 4, for example, the state consists of a tuple of four values. These represent an amount of cumulated GHG emissions, an implemented emission level, a level of availability of efficient technologies for reducing GHG emissions and a state of the world. In that problem, we will assume that the decision maker can only distinguish between low and high emissions

$$EmissionLevel = \{\, Low, High \,\}$$

and available or unavailable efficient GHG emission reduction technologies

$$Technology = \{\, Available, Unavailable \,\}$$

Similarly, the state of the world will be just good or bad:

$$World = \{\, Good, Bad \,\}$$

In realistic SEPs, decision makers typically have to select between more than two emission levels, efficient technologies for reducing GHG emission are available to certain degrees and the state of the world is slightly more multifaceted than just good or bad.

The second notion that characterizes a sequential emission process are the *controls* available to the decision maker. In the context of climate change studies, controls are often referred to as options, actions or policies. To avoid confusion with the notion of policy from section 2.3 below, we will call them controls.

In GHG emission problems, controls are often phrased in terms of abatement levels or, equivalently, in terms of maximum GHG emissions growth rates. Thus, for instance, in Webster (2000) and over the first decision step (for the time interval between 2010 and 2019) controls can be one of eight values: 0, 0.2, 0.4, 0.6, 0.8, 1.0, 1.2 and 1.4. Here, a value of 0.4 represents a maximal emissions growth rate of 0.4%. In the emission problem of section 4, we will further oversimplify this
picture and only consider low and high GHG emissions.

Notice that, in general, not all controls are available in every state and at every decision step. In other words, the abatement levels that can be selected in a given state can depend on that specific state. Thus, in our problem from section 4, we allow the probability of implementing low (high) emissions in the next period to depend on the current emission level. As discussed in Webster (2008), the probability of implementing low (high) emission levels in the next period is higher if the current emission
are already low (high) than if the current emissions are high (low). This kind of uncertainty account for, among others, the inertia of legislation and, of course, political instabilities. Thus, one can fully describe the states and the controls of a sequential decision process by defining two functions:

$$State \quad : (t : \mathbb{N}) \ \rightarrow \ Type$$
$$Control : (t : \mathbb{N}) \ \rightarrow \ (x : State \ t) \ \rightarrow \ Type$$

The interpretation is as follows: $State \ t$ denotes the set of states the decision maker can observe at the $t$-th decision step. Similarly, $Control \ t \ x$ are the controls that are available to the decision maker at decision step $t$ and in state $x$. Remember that, as explained in section 1.4, we denote function application by juxtaposition.

The third notion that characterizes a sequential emission process is that of a *transition function*. Informally, transition functions describe how states change, at each decision step, as a consequence of the controls selected by the decision maker. Thus,
in a deterministic decision process the transition function has the type

$$next : (t : \mathbb{N}) \ \rightarrow \ (x : State \ t) \ \rightarrow \ (y : Control \ t \ x) \ \rightarrow \ State \ (t + 1)$$

Again, the interpretation is that for every $t : \mathbb{N}$, $x : State \ t$ and $y : Control \ t \ x$, $next \ t \ x \ y$ is *the* new state at decision step $t + 1$. Notice that the time between two successive decisions does not need to be constant. In a time-dependent decision process, for instance, there could be a function

$$time : (t : \mathbb{N}) \ \rightarrow \ Real$$

with $time \ (t + 2) - time \ (t + 1) \neq time \ (t + 1) - time \ t$ for all (or, perhaps, only for certain) values of $t$. In Webster (2000), for instance, the author investigates two-steps decision problems in which the first period extends over 10 years and the second period extends over 80 years.

## 2.2   Sequential emission problems

A decision process becomes a decision problem when we fully specify the costs and the benefits that are associated with each transition. This can be done by defining a *reward* function. A reward function is a function that associates a value, at each decision step, to every current state, selected control and next state:

$$reward : (t : \mathbb{N}) \rightarrow (x : State\ t) \rightarrow (y : Control\ t\ x) \rightarrow (x' : State\ (t+1)) \rightarrow Val$$

As usual, we write *reward* in curried form and *reward t x y x'* : *Val* denotes the reward of selecting the control $y$ in $x$ at step $t$ and ending up in $x'$. Typically, *Val* is $\mathbb{R}$. An obvious question is: Why shall *reward* explicitly depend on $x'$? If $x'$ is *the* next state

$x' = next\ t\ x\ y$

it seems that $(t : \mathbb{N}) \rightarrow (x : State\ t) \rightarrow (y : Control\ t\ x) \rightarrow Val$ would be a more appropriate signature for *reward*. The reason for including *a* new state $x'$ in the signature of *reward* is uncertainty, as we explain in the following paragraphs. We have seen that, in deterministic decision processes, transition functions have the type

$$(t : \mathbb{N}) \rightarrow (x : State\ t) \rightarrow (y : Control\ t\ x) \rightarrow State\ (t+1)$$

What if the decision process is affected by uncertainties? If selecting an abatement level in a given state has uncertain outcomes (perhaps because of externalities or because the consequences of implementing certain emission reductions are not fully understood), it would be unsuitable to describe the decision process in terms of a transition function that returns a single next state. In this case, the transition function should return a set of *possible* next states or a probability distribution of next states. As detailed in Botta et al. (2017b, a), we can account for different kinds of uncertainties in decision processes with transition

functions of the form

$$next : (t : \mathbb{N}) \rightarrow (x : State\ t) \rightarrow (y : Control\ t\ x) \rightarrow M\ (State\ (t+1))$$

where $M$ is a functor. It represents the type of uncertainties underlying the decision process. For deterministic processes, $M$ is just the identity functor: $M = Id$. For stochastic processes, $M$ represents probability distributions. This is the case considered in this work. Thus, we take $M = Prob$ where $Prob\ X$ is the type of simple probability distributions[3] on $X$. Therefore, *next t x y*

is a probability distribution on next states that is, a value of type $Prob\ (State\ (t+1))$. The states in *next t x y* are those that can be obtained after decision step $t$ by selecting $y$ in state $x$. Thus, in a stochastic decision process, selecting a control does not yield a unique next state but a whole set of possible next states with their probabilities. Therefore, the reward function has to explicitly depend on $x'$ because this cannot be computed from the current state $x$ and the selected control $y$ unambiguously. This justifies the signature of *reward* as given above.

We can summarize the results obtained so far in the observation that stochastic sequential emission problems can be specified in terms of four functions:

$$
\begin{aligned}
State\ \ \ \ &: (t : \mathbb{N}) \rightarrow Type \\
Control &: (t : \mathbb{N}) \rightarrow (x : State\ t) \rightarrow Type \\
next\ \ \ \ &: (t : \mathbb{N}) \rightarrow (x : State\ t) \rightarrow (y : Control\ t\ x) \rightarrow Prob\ (State\ (t+1)) \\
reward\ &: (t : \mathbb{N}) \rightarrow (x : State\ t) \rightarrow (y : Control\ t\ x) \rightarrow (x' : State\ (t+1)) \rightarrow Val
\end{aligned}
$$

---

[3]In a nutshell, simple probability distributions are probability distributions with finite support, see Botta et al. (2017a).

We define these functions for our stylized emission problem in section 4. For the time being, we need to better understand the decision problem that four such functions specify. This is crucial for understanding the notions of policy and of policy sequence introduced in the next section.

The idea is that, for a fixed number of decision steps, the decision maker seeks controls (emission levels) that maximize a sum of the rewards obtained over those steps. The emphasis here is on "a sum": depending on the specific problem at stake, future rewards might need to be discounted and the way values of type $Val$ are added up might not be completely trivial. As explained in detail in Botta et al. (2017a), fully specifying stochastic SDPs requires defining $State$, $Control$, $next$, $reward$ and choosing a measure for weighting uncertain outcomes. Formally, a measure is just a function that reduces probability distribution on values to values

$$meas : Prob\ Val\ \rightarrow\ Val$$

The expected value function is probably the most widely used measure in the study of stochastic SDPs. But other measures are possible. Depending on the specific problem and on the kind of uncertainties, other measures might be more suitable than the expected value. Thus, for instance, a risk-averse decision maker might adopt a $worst$ measure rather than relying on the expected value. It is also conceivable, that a decision maker adopts different measures of uncertainty at different decision steps. The theory summarized in appendix A can be easily extended to cope with this situation. In section 4, we walk the reader through the full specification of our stylized emission problem, included uncertainty measures.

Solving SDPs is not trivial. For this, we instantiate the generic backward induction algorithms presented in Botta et al. (2017b, a). We do not need to discuss these methods in detail here but, before we move to section 4, it is important to achieve a good understanding of what it means to solve a stochastic SDP and of what it means for sequences of policies to be optimal.

In the rest of this section, we informally discuss the notions of policy, policy sequence and optimality of policy sequences. We do so in the context of sequential emission problems but the ideas apply to SDPs in general. In section 3, we discuss a number of basic facts about sequential emission problems. These, too, apply to sequential decision problems without loss of generality.

## 2.3 Emission policies

We have pointed out that, in stochastic sequential emission problems, selecting an emission (abatement) level at a given decision step and in a given state does not usually yield a unique next state. Instead, we obtain a probability distribution on next states. The distribution encodes the uncertainties associated with the decision process at study. Thus, for instance, the decision maker might select to reduce emissions by 2% but what actually gets implemented is a smaller reduction, perhaps because of political inertia or as a consequence of an increased economic activity.

One consequence of uncertainties is that, even if the decision maker could fix a priori an emission schedule or path[4], she would not know the state obtained after a fixed number of decision steps. This is, again, because each single step yields a probability distribution on next states, not a single next state.

---

[4]Strictly speaking, this is impossible because, as we have seen, what are feasible emissions in a given state may depend on that state.

Thus, the best a decision maker can hope to obtain as a solution of a stochastic sequential emission problem is a sequence of rules that tell her which control (abatement level) to select for each decision step and, at that step, for each possible state.

In control theory, such "rules of action" are called policies. This is also the sense in which the word policy has been used in Botta et al. (2017b, a). The control theoretical notion matches quite well the notion of strategy in game theory (Fudenberg and Tirole, 1991), but notice that, in plain English, the term policy is ambiguous: sometimes it is used to denote a plan (course) of action, sometimes a rule of action, see www.merriam-webster.com/dictionary/policy.

Here we follow the control theory standard and policy sequences are just sequences of functions, one for each decision step. A sequence of policies for $n + 1$ decision steps consists of a policy $p$ for the $t$-th decision step and of a policy sequence $ps$ for further $n$ steps. Formally we write

$$(p :: ps) : PolicySeq \ t \ (n + 1)$$

with $p \ : \ Policy \ t \ (n + 1)$ and $ps \ : \ PolicySeq \ (t + 1) \ n$. Here, :: is the operator that prepends a policy to a (possibly empty) policy sequence, see appendix A and sections 3.5, 3.7 and 3.9 of Botta et al. (2017a). More formally, if $ps = [p1, p2, p3]$ then $p :: ps = [p, p1, p2, p3]$ for all $p \ : \ Policy \ t \ (n + 1)$, $ps \ : \ PolicySeq \ (t + 1) \ n$.

But what does it mean for a sequence of emission policies to be optimal? The decision maker aims at maximizing the sum of rewards over a fixed number of steps. Thus, $(p :: ps)$ is *an* optimal policy sequence for $n + 1$ decision steps iff no other sequence attains a higher sum of rewards (over $n + 1$ steps) for any given $x \ : \ State \ t$.

While fairly intuitive, formalizing this notion of optimality is not completely trivial. This is because, in a stochastic emission problem, a selected abatement level does not entail a unique next state, as explained above. Thus, for any *possible* next state (and, therefore, for any possible value of taking $n$ further decision steps taken with the policies of $ps$ and starting from that state) we have a corresponding reward and a probability. Such a probability distribution of rewards has to be measured with *meas* in order to obtain the value of making $n + 1$ decision steps according to the policy $p$ and to the policy sequence $ps$.

In appendix A, we discuss the computation of the value of policy sequences in detail. In order to get an intuition of the notion of optimality for policy sequences, it is sufficient to recognize that one can precisely define a function

$$val \ : \ (x \ : \ State \ t) \ \rightarrow \ PolicySeq \ t \ n \ \rightarrow \ Val$$

In the theory of SDPs, *val* is called the value function. As one would expect, *val x ps* is the value, in terms of the measured sum of possible rewards, of performing $n$ decision steps with the policy sequence $ps$ and starting in state $x$. Crucially, *val x ps* only depends on $State$, $Ctrl$, $next$, $reward$, $meas$ and on the rule for adding up rewards.

The value function allows us to give a precise meaning to the intuitive notion of optimality of policy sequences discussed above. More importantly, it allows us to actually compute optimal sequences of policies, at least for decision problems that fulfill certain natural conditions.

Again, a comprehensive discussion of the notion of optimality and of the conditions under which optimal policy sequences can be computed goes well beyond the scope of this work. We refer the interested reader to appendix A and to Botta et al. (2017a) and close this section by recalling an often neglected fact on decision making under uncertainty.

A fundamental difference between decision making under deterministic transition functions and decision making under uncertainty is that, in the latter case, regret cannot, in general, be avoided. Here, by regret we mean a judgment in hindsight, often triggered by an unlucky sequence of transitions. Thus, for instance, a system for optimal routing may recommend a driver to leave a highway in order to avoid an upcoming traffic jam. On the alternative road, the driver may get involved in a car accident and finally regret having left the highway. Of course, the driver's regret does not change the fact that leaving the highway was a best choice (under the problem's reward function, measure of possible rewards, etc.) at the point in time in which she had to make her choice.

In both the deterministic and in the uncertain case, the notion of "best decision" is the same: at, say, decision step $t$ and in $x : State\ t$, a best decision $y^* : Ctrl\ t\ x$ is a decision that cannot be bettered (in terms of sum of possible rewards) given the decision problem (that is, the functions $State$, $Ctrl$, $next$, $reward$, the measure $meas$, and the rule for adding rewards) and a sequence of policies (optimal or not) for taking $n$ further decisions.

But when the outcome of a decision step is a probability distribution on next states, we will have many possible trajectories of length $n+1$ starting in $x$ instead of just one. In general, there is nothing preventing some of these trajectories to contain states that make any best decision in $x$ regrettable. This is true even for trajectories of length 1 that is, for $n = 0$.

## 3  Logical consequences of SEPs

In this section we discuss some logical consequences of the notions introduced in 2. A first consequence of the notion of optimal policy sequence is that optimal decisions may vary over time: a best control at a given step does not need to be a best control at a subsequent (or previous) step even if the decision maker observes the same state at both decision steps. There is nothing worrying with this fact: time-inconsistency of optimal policies and Bellman's principle of optimality (Bellman, 1957) are perfectly consistent!

Another consequence of the notions introduced in section 2 is that exploiting available information is crucial in decision making under uncertainty. We have seen that, under uncertainty, regret cannot in general be avoided. In spite of this fact, the notion of optimal policy sequence and of "best" decision are both clear and compelling: optimal policy sequences for SEPs provide decision makers with rules for selecting emission levels that, at any decision step, cannot be bettered given the information available to the decision maker at that step.

The crucial point is exploiting the information available at a given decision step. As seen in section 2, this information is coded in the notion of $State$ and the mechanism for exploiting such information are policies or action rules. Taking decisions on the basis of optimal policies is in most cases better than selecting controls according to fixed (ex-ante) action plans. This is because, in contrast to fixed action plans, policies provide an action for every possible state that can eventually be reached (ex-post) at a given decision step. They account for all the information available to the decision maker at that step. Further, optimal policies entail actions that cannot be bettered.

In section 5, we discuss optimal policies for the emission problem of section 4. Because these policies are computed using the verified framework presented in Botta et al. (2017b, a), we know (in spite of the uncertainties affecting emission problems,

for certain!) that the conclusions that we draw for our uncertain emission problem are logical consequences of the problem specification. Computing optimal policies with a verified implementation is crucial because, in contrast to other properties of solutions of computational problems, optimality cannot in general be established by testing. This is a well know case in which proving is (albeit difficult, still) easier than testing, see Ionescu and Jansson (2013).

A third obvious logical consequence of the notions introduced in the previous section is that best controls and optimal policies are not, in general, unique. In section 5 we discuss a problem setup in which both increasing and decreasing emissions is optimal. When applying optimal control to inform policy advice and decision making is important to keep in mind that optimal policies are not necessarily unique: different optimal emission sequences can yield different sets of possible emission paths. Decision makers might not be able to distinguish them in terms of measures of possible sums of rewards, but they still

might have reasons to prefer certain optimal emission policies to others. For instance, precautionary approaches might lead decision makers to prefer optimal policies that entail low risk levels to high risk optimal policies.

     Another logical consequence of decision making under uncertainty is that the value of policies depends not only on the problem-specific reward function and on the way rewards are added (e.g. via discounting) but also on how the decision maker weighs uncertain outcomes. This is captured by the measure function $meas$. Different measures reflect different attitudes or

dispositions, e.g., towards risk.

     As explained in Ionescu (2009), decision makers are free to choose whatever measure they like as far as it fulfills a monotonicity condition. Informally, this condition says that if one increases the $Val$-values of a probability distribution by any arbitrary amount (by letting their probabilities unchanged), its measure shall not decrease, see appendix A. The expected value in much the same way as worst and best case measures fulfill this condition. But notice that, as discovered in Ionescu (2009)

in the context of a formalization of vulnerability notions (see pp. 112-116), measures that pick up the most (least) probable $Val$-value of a probability distribution do violate the monotonicity condition. It is a responsibility of scientific advisors to make sure that decision making is informed by meaningful, monotonic measures.

## 4    A stylized sequential emission problem

In this and in the next section, we study how optimal sequences of GHG emission policies are affected by:

1. Uncertainty about the implementability of decisions on GHG emission reductions.

     2. Uncertainty about the availability of efficient technologies for reducing GHG emissions.

     3. Uncertainty about the implications of exceeding a critical threshold of cumulated GHG emissions.

As anticipated in the introduction, we first specify a stylized sequential emission problem that accounts for all three sources of uncertainty and yet is simple enough to support investigating the logical consequences of different assumptions on such

uncertainties. In section 5 we discuss the optimal policies obtained for our stylized problem under different assumptions.

     We specify our stylized emission problem by instantiating the theory for SDPs summarized in appendix A. Technically, this is done by defining all the undefined variables in the modules that implement the theory. For the implementation provided

in the `SequentialDecisionProblems` component of Botta (2016–2017), these are the undefined variables (holes) in `CoreTheory`, `FullTheory` and in the ancillary modules `Utils`, `CoreTheoryOptDefaults`, `FullTheoryOptDefaults`, `FastStochasticDefaults`, `TabBackwardsInduction` and `TabBackwardsInductionOptDefaults`. For a detailed discussion on how to specify a SDP, see Botta et al. (2017a).

In the rest of this section, we skip most technical details and focus on the specification of the emission problem from an applicational perspective. A complete implementation of our specification is available in `applications/EmissionGame2`. This is a subcomponent of `SequentialDecisionProblems` in Botta (2016–2017).

As anticipated in the introduction, we specify our stylized emission problem as a stochastic SDP. Thus, $M = Prob$. We have to define the four functions $State$, $Control$, $next$ and $reward$ introduced in section 2. We start by defining the controls, that is

the options available to the decision maker.

## 4.1    Controls

In our stylized emission problem, at each decision step, the decision maker can only select between low and high GHG emissions. Thus,

$Control\ t\ x = LowHigh$

where $LowHigh$ is a type inhabited by only two values: $Low$ and $High$. The idea is that low emissions, if actually implemented, increase the cumulated GHG emissions less than high emissions.

## 4.2    States

At each decision step, the decision maker has to choose between low and high emission levels on the basis of four data: a measure of cumulated GHG emissions, the current emission level (itself either low or high), the availability of effective

technologies for reducing GHG emissions and a "state of the world". Effective technologies for reducing GHG emissions can be either available or unavailable. The state of the world can be either good or bad:

$State\ t = (CumulatedEmissions\ t, LowHigh, AvailableUnavailable, GoodBad)$

The idea is that the decision process starts with zero cumulated emissions, high emission levels, unavailable GHG technologies and with the world in a good state. In these conditions, the probability for the world to turn to the bad state is low. But if the

cumulated emissions increase beyond a fixed critical threshold, the probability that the world becomes bad increases. If the world is in the bad state, there is no chance to come back to the good state. Similarly, the probability that effective technologies for reducing GHG emissions become available increases after a fixed number of decision steps. Once available, effective technologies stay available for ever.

In a realistic problem, the capability of actually implementing a decision on a given GHG emission level typically depends

on a variety of factors. In our stylized problem, we follow Webster (2000, 2008) and focus on the uncertainties about the implementability of decisions on GHG emission reductions that come from inertia: implementing low emissions is easier

when low emission measures are already in place than when the current emissions are high. Similarly, implementing high emission measures is easier if the current emissions are high than under low emissions regulations.

## 4.3 Transition function

We have defined $State\ t$ to be a tuple of values representing cumulated GHG emissions, the current emission level, the availability of effective technologies for reducing GHG emissions and the state of the world at decision step $t$. As our stylized emission problem is stochastic, its transition function at decision step $t$ yields a probability distribution on values of type $State\ (t+1)$.

The idea is that low emission levels leave the cumulated emissions unchanged and high emissions increase the cumulated emissions. Without loss of generality, we can take such increase to be one. We have mentioned that the probability of the state of the world to become bad depends on a critical cumulated emissions threshold. Let's call this threshold $crE$

$crE\ :\ Double$

and let $pS1$ and $pS2$ the probabilities of staying in a good world when the cumulated emissions are smaller or equal to $crE$ and greater than $crE$, respectively:

$pS1\ :\ NonNegDouble$
$pS2\ :\ NonNegDouble$

Thus, the probabilities of getting into a bad world below and above the threshold are $1 - pS1$ and $1 - pS2$, respectively. As a sanity check, we require $pS2$ to be less or equal to $pS1$.

Next, we have to specify the uncertainties about the availability of efficient technologies for reducing GHG emissions. This, too, can be done in terms of a critical number of decision steps

$crN\ :\ \mathbb{N}$

and of two probabilities: the probability of effective technologies for reducing GHG emissions becoming available when the number of decision steps is below or at $crN$ and the probability for the case in which $t$ is above $crN$:

$pA1\ :\ NonNegDouble$
$pA2\ :\ NonNegDouble$

Also for these probabilities we need a sanity check: $pA1$ shall be at most equal to $pA2$. Finally, we have to specify the uncertainties about the implementability of decisions on GHG emission reductions. Following the discussion in the previous section, we do so in terms of four conditional probabilities. These are the probability of implementing low emission measures when the current emissions measures are low and low emissions are selected $pLL$, the probability of implementing low emission measures when the current emissions measures are high and low emissions are selected $pLH$ and their counterparts for high emissions:

$$pLL \ : \ NonNegDouble$$
$$pLH \ : \ NonNegDouble$$
$$pHL \ : \ NonNegDouble$$
$$pHH \ : \ NonNegDouble$$

Also for these probabilities, we require two sanity checks to be fulfilled: $pLH$ shall not exceed $pLL$ and $pHL$ shall not exceed $pHH$. With these parameters in place, the transition function $next$ can be implemented by cases. For a full implementation, we refer the reader to `applications/EmissionGame2`. As an example we discuss here the case in which the current state is

$$x = (e, High, Unavailable, Good)$$

the decision maker has opted for low emissions, $e$ is smaller or equal to $crE$ and $t$ is smaller or equal to $crN$. In this case, the
result of $next \ t \ x \ Low$ is a probability distribution with the following assignments:

$$
\begin{array}{llll}
(e, & Low, Unavailable, Good) \Rightarrow & pLH \ * (one - pA1) \ * & pS1 \\
(e+1, High, Unavailable, Good) \Rightarrow & (one - pLH) * (one - pA1) \ * & pS1 \\
(e, & Low, \ \ Available, \ \ Good) \Rightarrow & pLH \ * & pA1 \ * & pS1 \\
(e+1, High, \ \ Available, \ \ Good) \Rightarrow & (one - pLH) \ * & pA1 \ * & pS1 \\
(e, & Low, Unavailable, Bad) \Rightarrow & pLH \ * (one - pA1) * (one - pS1) \\
(e+1, High, Unavailable, Bad) \Rightarrow & (one - pLH) * (one - pA1) * (one - pS1) \\
(e, & Low, \ \ Available, \ \ Bad) \Rightarrow & pLH \ * & pA1 \ * (one - pS1) \\
(e+1, High, \ \ Available, \ \ Bad) \Rightarrow & (one - pLH) \ * & pA1 \ * (one - pS1)
\end{array}
$$

Similarly for the other cases. Notice that the marginal probability of the new state to enter a bad world is $one - pS1$, as one
would expect. Similarly, the probability of effective technologies for reducing GHG emissions becoming available is $pA1$ (we are considering the case $t \leqslant crN$) and the probability of implementing low emission measures is $pLH$ as the current emission levels are high.

## 4.4   Reward function

To complete the specification of our stylized emission problem, we have to define the reward function and the measure

$$meas \ : \ Prob \ Val \ \to \ Val$$

according to which the decision maker weights uncertain outcomes. Unless stated otherwise, we will take $Val$ to be $NonNegDouble$ (non-negative double precision floating point numbers) and $meas$ to be the expected value function. In this section we focus the attention on the reward function

$$reward \ : (t \ : \ \mathbb{N}) \ \to \ (x \ : \ State \ t) \ \to \ (y \ : \ Control \ t \ x) \ \to \ (x' \ : \ State \ (t+1)) \ \to \ Val$$

The idea is that being in a good world yields one unit of benefits per step and being in a bad world yields less benefits. We can formalize this idea by introducing a dimensionless number

$$badOverGood : NonNegDouble$$

which represents the ratio between the step benefits in a bad world and the step benefits in a good world. It goes without saying that a constant ratio is a very crude approximation that can only be justified in a stylized problem. In sequential emission problems aiming at informing decision making under realistic conditions, the costs and the benefits of not transgressing global emission thresholds are likely to be time dependent and have to be carefully estimated, e.g., by running global climate models coupled with economic models and perhaps energy models. Unless otherwise stated, we will take $badOverGood$ to be equal to $0.5$. Of course, we require the $badOverGood$ ratio to be smaller or equal to one.

Emitting GHGs also brings step benefits, e.g. by supporting economic growth. These can be represented as a fraction of the step benefits of being in a good world. Moreover, low emissions bring less benefits (higher costs) than high emissions and reducing emissions when effective technologies are unavailable costs more than reducing emissions when such technologies are available. We can summarize this state of affairs in terms of three dimensionless numbers. A first number represents the ratio between the step benefits of low emissions and the step benefits in a good world when effective technologies for reducing GHG emissions are unavailable

$$lowOverGoodUnavailable : NonNegDouble$$

A second number represents the same ratio when effective technologies are available

$$lowOverGoodAvailable : NonNegDouble$$

and, finally, the ratio between the step benefits obtained through high emissions and the step benefits in good worlds

$$highOverGood : NonNegDouble$$

We require both $lowOverGoodUnavailable$, $lowOverGoodAvailable$ and $highOverGood$ to be smaller or equal to one, $lowOverGoodUnavailable$ to be smaller or equal to $lowOverGoodAvailable$ and the latter to be smaller or equal to $highOverGood$. With these notions in place, we can easily implement the reward function of our stylized emission problem. The idea is that the rewards only depend on the next state (the state during the period starting with the current decision) not on the current state or on the selected control. We have 8 cases with the following assignments

$$
\begin{aligned}
(e, High, Unavailable, Good) &\Rightarrow one && + one * highOverGood \\
(e, High, Unavailable, Bad) &\Rightarrow one * badOverGood + one * highOverGood \\
(e, High, \quad Available, \; Good) &\Rightarrow one && + one * highOverGood \\
(e, High, \quad Available, \; Bad) &\Rightarrow one * badOverGood + one * highOverGood \\
(e, Low, Unavailable, Good) &\Rightarrow one && + one * lowOverGoodUnavailable \\
(e, Low, Unavailable, Bad) &\Rightarrow one * badOverGood + one * lowOverGoodUnavailable \\
(e, Low, \quad Available, \; Good) &\Rightarrow one && + one * lowOverGoodAvailable \\
(e, Low, \quad Available, \; Bad) &\Rightarrow one * badOverGood + one * lowOverGoodAvailable
\end{aligned}
$$

| parameter | value | constraints |
|---|---|---|
| *badOverGood* | 0.5 | $badOverGood \leqslant 1$ |
| *highOverGood* | 0.3 | $highOverGood \leqslant 1$ |
| *lowOverGoodAvailable* | 0.2 | $lowOverGoodAvailable \leqslant highOverGood$ |
| *lowOverGoodUnavailable* | 0.1 | $lowOverGoodUnavailable \leqslant lowOverGoodAvailable$ |

**Table 1.** Reward function: parameters, default values and sanity constraints.

In summary, the parameters that define the reward function of our stylized emission problem, their default values and sanity constraints are:

Completing the specification of our problem and computing optimal sequences of emission policies requires filling in some more details. These are annotated and discussed in `applications/EmissionGame2`. They are pertinent to the notions of reachability, viability, finiteness and decidability. These notions are crucial for understanding the problem of computing optimal policies under uncertainty but their discussion would go well beyond the scope of this work. We refer the interested reader to Botta et al. (2017a).

## 5   Optimal policies

In this section we discuss optimal emission policies for the stylized emission problem of section 4 and study the impact of the uncertainties (1)–(3) on such policies. As explained in section 3, the computed policies have been machine-checked to be optimal. Thus, they only depend on our problem specification. This is simple enough to allow deducing some general properties that optimal decisions — decisions taken according to optimal policy sequences — have to fulfill.

A first one is that no optimal policy sequence can require selecting low emissions when the state of the world is bad. This is because, as posited in section 4, there is no way to make a transition from a bad world to a good world and, in a bad world in much the same way as in a good world, higher emissions bring more emission benefits. In other words, reducing emissions can only pay off if it makes it possible (albeit not certain) to avoid transitions to a bad world, if perhaps only for a limited number of steps. Once such a transition has taken place, reducing emissions is pointless. A consequence of this matter of facts is that in the last step it is always optimal to select high emissions. In a realistic emission problem, one could easily prevent this situation by introducing a suitable "unsustainability" penalty in the reward function at the last decision step.

We do not need to deal with such complications here but it is perhaps useful to point out that very often, seemingly natural and innocuous assumptions (in this case, that the number of decision steps is finite and known to the decision maker) can have non-trivial consequences on "best" decisions. Thus, for instance, the rate at which rewards are discounted in integrated assessment models of climate change typically has a severe impact on optimal emission policies. Thus, in policy advice, it is crucial to apply theories that require all assumptions to be made explicitly. This was one of the guiding criteria in developing the theory of policy advice and avoidability discussed in Botta et al. (2017a).

Unless specified, we consider 9 decision steps with $crE = 4$ and $crN = 2$. Thus, it takes at least 5 decision steps (and 5 periods with high emissions) to achieve states in which the sum of the cumulated emissions exceeds $crE$ and, therefore, the probability of a transition to a bad world increases from $pS1$ to $pS2$. Similarly, with $crN = 2$, it takes 3 decision steps to achieve states in which the probability that effective technologies for reducing GHG emissions become available increases

from $pA1$ to $pA2$.

In other worlds, if $pS1 = pA1 = 0$ and $pS2 = pA2 = 1$, effective technologies will be available (with certainty) after 4 decision steps. And after 5 periods of high emissions, a transition to a bad world will occur. This is the deterministic base case studied in the next section.

## 5.1 The deterministic base case

Before studying the impact of uncertainties on optimal policies, we consider the certain case. Beside $pS1 = pA1 = 0$ and $pS2 = pA2 = 1$ we also have $pLL = pLH = pHL = pHH = 1$. Thus, there is no uncertainty about the implementability of emission measures: decisions of reducing or increasing emissions are implemented with probability one.

Notice that the absence of whatsoever uncertainties implies that, for any initial state and policy sequence (optimal or not) there is exactly one possible state-control trajectory. Namely that determined by that policy sequence. Thus, for instance, if

we start in (0,H,U,G) (zero cumulated emissions, high emissions, unavailable efficient technologies and a good world) and adopt the policy of constantly increasing emissions, we obtain the state-control trajectory

```
[((0,H,U,G),H), ((1,H,U,G),H), ((2,H,U,G),H), ((3,H,U,G),H), ((4,H,A,G),H),
 ((5,H,A,G),H), ((6,H,A,B),H), ((7,H,A,B),H), ((8,H,A,B),H), ((9,H,A,B), )]
```

with probability one. The sum of rewards associated to this "certain" trajectory is 9.7: these result from five periods in a good world (step benefits equal to one), 4 periods in a bad world (step benefits 0.5) and 9 periods of high emissions (emission benefits per step of 0.3). As expected, efficient technologies for reducing GHG emissions become available at decision step 4

(after 4 decisions) and the transition to a bad world takes place after 5 periods of high emissions and 6 decisions. We can do a little bit better by selecting low emissions at every step. In this case the state-control trajectory is

```
[((0,H,U,G),L), ((0,L,U,G),L), ((0,L,U,G),L), ((0,L,U,G),L), ((0,L,A,G),L),
 ((0,L,A,G),L), ((0,L,A,G),L), ((0,L,A,G),L), ((0,L,A,G),L), ((0,L,A,G), )]
```

What are optimal policy sequences like in the certain case? The intuition is that, in at least 4 decision steps, emissions should be high. This yields higher rewards at no risk of getting into a bad world. One would also expect that lower emissions are selected (and implemented with certainty) in states in which efficient technologies for reducing GHG emissions are available.

The trajectory associated with an optimal sequence of policies

```
[((0,H,U,G),H), ((1,H,U,G),H), ((2,H,U,G),H), ((3,H,U,G),H), ((4,H,A,G),L),
 ((4,L,A,G),L), ((4,L,A,G),L), ((4,L,A,G),L), ((4,L,A,G),H), ((5,H,A,G)  )]
```

shows that such intuition is correct. The sum of rewards associated to this trajectory is 11.3. By selecting low emission starting from the fifth decision step, the optimal policy guarantees that the world stays in the good state. At the last decision step, high emissions are selected, as anticipated.

The computation supports the intuition that, in a world without uncertainties, it is best delaying emission reductions until efficient technologies become available. Of course, this requires knowing the critical number of decision steps $crN$.

## 5.2 The impact of uncertainties about the implementability of decisions on emission reductions.

What happens to optimal policies if we factor in uncertainties about the implementability of decisions on emission reductions or increases?

Let's consider the case in which the probability of implementing low emission measures in the next period is higher if the current emissions are already low than in the case in which the current emissions are high. Conversely, the probability of implementing high emission in the next period is higher if the current emissions are high. In other words, we have $pLH < pLL$ and $pHL < pHH$ instead of $pLL = pLH = pHL = pHH = 1$. Specifically, consider optimal policies for the case $pLL = pHH = 0.9$ and $pLH = pHL = 0.7$.

Our decision problem is now not anymore deterministic. Thus, a policy (optimal or not) entails a whole set of possible future state-control trajectories. More precisely, we have $2^9 = 512$ possible trajectories: we take 9 decision step and, at every decision step and no matter whether we select low or high emissions, we have two possible outcomes. Now, the "business as usual" policy of always selecting high emissions yields the trajectory

```
[((0,H,U,G),H), ((1,H,U,G),H), ((2,H,U,G),H), ((3,H,U,G),H), ((4,H,A,G),H),
 ((5,H,A,G),H), ((6,H,A,B),H), ((7,H,A,B),H), ((8,H,A,B),H), ((9,H,A,B), )]
```

with probability $0.9^9 \approx 0.387$. The two next most likely trajectories are

```
[((0,H,U,G),H), ((1,H,U,G),H), ((2,H,U,G),H), ((3,H,U,G),H), ((4,H,A,G),H),
 ((5,H,A,G),H), ((6,H,A,B),H), ((7,H,A,B),H), ((8,H,A,B),H), ((8,L,A,B)  )]
```

and

```
[((0,H,U,G),H), ((1,H,U,G),H), ((2,H,U,G),H), ((3,H,U,G),H), ((4,H,A,G),H),
 ((4,L,A,G),H), ((5,H,A,G),H), ((6,H,A,B),H), ((7,H,A,B),H), ((8,H,A,B), )]
```

with probabilities of $0.043$ and $0.033$. The expected sum of rewards (remember that $meas$ is the expected value function) is $9.904$. The computed optimal policies for the same problem yield the trajectory

```
[((0,H,U,G),H), ((1,H,U,G),H), ((2,H,U,G),L), ((2,L,U,G),L), ((2,L,A,G),L),
 ((2,L,A,G),L), ((2,L,A,G),H), ((3,H,A,G),H), ((4,H,A,G),H), ((5,H,A,G), )]
```

with probability $0.234$. The two next most likely trajectories are

```
[((0,H,U,G),H), ((1,H,U,G),H), ((2,H,U,G),L), ((3,H,U,G),L), ((3,L,A,G),L),
 ((3,L,A,G),L), ((3,L,A,G),L), ((3,L,A,G),H), ((4,H,A,G),H), ((5,H,A,G), )]
```

```
[((0,H,U,G),H), ((1,H,U,G),H), ((2,H,U,G),L), ((2,L,U,G),L), ((2,L,A,G),L),
 ((2,L,A,G),L), ((2,L,A,G),H), ((2,L,A,G),H), ((3,H,A,G),H), ((4,H,A,G), )]
```

both with probability 0.078. The expected sum of rewards (remember that $meas$ is the expected value function) is 11.085. Notice that, under uncertainties about the implementability of decisions on emission reductions or increases, optimal policies dictate more precautious best decisions: instead of waiting for efficient technologies for reducing GHG emissions to become available, optimal decision making requires starting to reduce emissions after only two decision steps.

The fact that higher uncertainties about the implementability of decisions on emission reductions or increases lead to more precautionary optimal policies is confirmed by computing optimal policies for the case $pLL = pHH = 0.7$ and $pLH = pHL = 0.5$. In this case optimal policies dictate low emissions in the first decision steps for the three most likely possible trajectories. This is still true in the limit $pLL = pHH = 0.5 + \epsilon$ for $\epsilon > 0$, $\epsilon \longrightarrow 0$ although the advantage of optimal policies against non-optimal policies (e.g. business as usual policies) in terms of expected rewards tends to zero as $\epsilon$ goes to zero.

In the limit case in which the decision maker has no power to enforce its emission decisions for the next period and $pLL = pHH = pLH = pHL = 0.5$, any policy sequence is optimal, as one would expect. As discussed in section 3, this is an example of non-uniqueness of optimal policies.

## 5.3    The impact of uncertainties about the availability of efficient technologies for reducing GHG emissions

What if the probability of efficient technologies becoming available after 3 decision steps is less than one and there is a small
but not zero probability that such technologies become available before 3 decision steps?

With the same uncertainties as in 5.2 ($pLL = pHH = 0.9$ and $pLH = pHL = 0.7$) and $pA1$, $pA2$ equal to 0.1 and 0.9 instead of 0 and 1, we have now $2^n * (n+1)$ possible trajectories[5] for $n$ decision steps. Thus, for $n = 9$, we have 5120 trajectories instead of just 512. The "business as usual" policy of always selecting high emissions yields the same most likely trajectory and a slightly higher expected sum of rewards: 9.91. The computed optimal policies also yield the same most likely trajectories
as in 5.2 although with lower probabilities, of course. The expected sum of rewards is 11.102.

Thus, perhaps surprisingly, uncertainties on the availability of efficient technologies for reducing GHG emissions have little impact on optimal decisions, at least when compared to the impact of uncertainties about the implementability of decisions on emission reductions.

---

[5]At each decision step, a possible state in which efficient technologies are not available, say a U-state, entails 4 possible next states: two in which efficient technologies are available and two in which they are not. A possible state in which efficient technologies are available (an A-state) only entails 2 possible next states because once technologies become available they stay available in all possible future states. Thus, after one decision step, we have two possible U-states and two possible A-states. After two decision steps, we have four possible U-states and eight possible A-states. After three decision steps we have eight possible U-states and twenty-four possible A-states. And so on.

## 5.4 The impact of uncertainties about the implications of exceeding a critical threshold of cumulated GHG emissions.

So far we have assumed that, if the critical cumulated GHG emissions threshold $crE$ was exceeded, the world would turn to a bad state with probability one. Conversely, for cumulated emissions below the $crE$, the probability of a transition into a bad world was zero.

What if we assume a 10% probability of turning to a bad world for cumulated emissions below the $crE$ and a 10% chance of staying in a good world above the critical threshold?

Adding these uncertainties to the certain "base" case yield 10 possible trajectories. These correspond to transitions to a bad world in the first, second, . . . and ninth decision step. In this scenario, always selecting high emissions yields the trajectory of the certain case

```
[((0,H,U,G),H), ((1,H,U,G),H), ((2,H,U,G),H), ((3,H,U,G),H), ((4,H,A,G),H),
 ((5,H,A,G),H), ((6,H,A,B),H), ((7,H,A,B),H), ((8,H,A,B),H), ((9,H,A,B), )]
```

with probability 0.531. The expected sum of possible rewards is lower than in the certain case: 9.076. Similarly, optimal policies under uncertainty about the implications of exceeding $crE$ yield the possible trajectory

```
[((0,H,U,G),H), ((1,H,U,G),H), ((2,H,U,G),H), ((3,H,U,G),H), ((4,H,A,G),L),
 ((4,L,A,G),L), ((4,L,A,G),L), ((4,L,A,G),L), ((4,L,A,G),H), ((5,H,A,G), )]
```

with probability 0.387. In the certain case, this was also "the" (certain) optimal trajectory. The expected sum of possible rewards is 9.731: much lower than in the certain case but still better than for the "business as usual" policies.

These results suggest that, as for the case of uncertainties about the availability of efficient technologies, uncertainties about the implications of exceeding $crE$ do not affect optimal policies substantially: the intuition that lower emissions should be selected (and implemented with certainty) in states in which efficient technologies for reducing GHG emissions are available still holds.

Adding uncertainties about the implications of exceeding $crE$ on the top of uncertainties about the implementability of decisions and of uncertainties about the availability of efficient technologies also does not change substantially the understanding obtained in section 5.2 and 5.3. But it brings some new unexpected results.

With $pLL = pHH = 0.9$, $pLH = pHL = 0.7$, $pA1 = 0.1$, $pA2 = 0.9$ and $pS1 = 0.9$, $pS2 = 0.1$ one obtains 51200 possible trajectories. For "business as usual" policies, the most likely is the usual

```
[((0,H,U,G),H), ((1,H,U,G),H), ((2,H,U,G),H), ((3,H,U,G),H), ((4,H,A,G),H),
 ((5,H,A,G),H), ((6,H,A,B),H), ((7,H,A,B),H), ((8,H,A,B),H), ((9,H,A,B), )]
```

with probability 0.135. Remember that, in absence of uncertainty about the implications of exceeding $crE$ the three most likely trajectories were

```
[((0,H,U,G),H), ((1,H,U,G),H), ((2,H,U,G),L), ((2,L,U,G),L), ((2,L,A,G),L),
 ((2,L,A,G),L), ((2,L,A,G),H), ((3,H,A,G),H), ((4,H,A,G),H), ((5,H,A,G), )]
```

```
[((0,H,U,G),H), ((1,H,U,G),H), ((2,H,U,G),L), ((3,H,U,G),L), ((3,L,A,G),L),
 ((3,L,A,G),L), ((3,L,A,G),L), ((3,L,A,G),H), ((4,H,A,G),H), ((5,H,A,G), )]

[((0,H,U,G),H), ((1,H,U,G),H), ((2,H,U,G),L), ((2,L,U,G),L), ((2,L,A,G),L),
 ((2,L,A,G),L), ((2,L,A,G),H), ((2,L,A,G),H), ((3,H,A,G),H), ((4,H,A,G), )]
```

with associated rewards 11.2, 11.3, 11.1 and probabilities 0.154, 0.051 and 0.051. The expected sum of possible rewards was 11.102. Adding 10% of uncertainty about the implications of exceeding $crE$ yields

```
[((0,H,U,G),H), ((1,H,U,G),H), ((2,H,U,G),H), ((3,H,U,G),L), ((3,L,A,G),L),
 ((3,L,A,G),L), ((3,L,A,G),L), ((3,L,A,G),H), ((4,H,A,G),H), ((5,H,A,G),)]

[((0,H,U,G),H), ((1,H,U,B),H), ((2,H,U,B),H), ((3,H,U,B),H), ((4,H,A,B),H),
 ((5,H,A,B),H), ((6,H,A,B),H), ((7,H,A,B),H), ((8,H,A,B),H), ((9,H,A,B), )]

[((0,H,U,G),H), ((1,H,U,G),H), ((2,H,U,B),H), ((3,H,U,B),H), ((4,H,A,B),H),
 ((5,H,A,B),H), ((6,H,A,B),H), ((7,H,A,B),H), ((8,H,A,B),H), ((9,H,A,B), )]
```

with expected rewards 11.3, 7.2, 7.7 and probabilities 0.059, 0.025 and 0.023, respectively. The expected sum of possible rewards is 9.543. Now, optimal policies for the most likely trajectory require postponing emission reductions by one step: low emission are selected starting from $t = 3$ instead of $t = 2$.

Notice that the optimal policies require constant high emissions both for the second and for the third most likely trajectories! This is because, in these trajectories, the world enters a bad state right after the first decision step (second trajectory) or after the second decision step (third trajectory). Indeed, the rewards associated to the second and to the third trajectories (7.2 and 7.7, respectively) are significantly lower than the rewards associated to the most likely trajectory (11.3).

Notice also that, even though the probability of transitions into a bad world is only 0.1 for cumulated emissions below $crE$, the trajectory that entails such a transition immediately after the first decision step (the second one) is more likely to occur than the trajectory in which the world stays in the good state for the first period (third one).

This seems at the first sight counter-intuitive. But it can easily be verified by inspection[6] and is in fact easily explained: the crucial point is that the probability of entering a bad world at the first decision step (and then, necessarily, staying in a bad world) is 0.1. By contrast, the probability of staying in a good world for one period and then getting into a bad world is, ceteris paribus, 0.9 * 0.1. This difference makes the second trajectory more likely than the third one. Of course, both trajectories are much less likely than the first one as in the cases discussed in 5.2 and 5.3.

---

[6]Given the probabilities $pS1$, $pS2$, $pA1$, $pA2$, $pLL$, $pLH$, $pHL$ and $pHH$ as above, the probability of a given trajectory is just the product of the probabilities of the corresponding transitions.

# 6 Conclusions

We have studied the impact of uncertainties about 1) the implementability of decisions on emission reductions, 2) the availability of technologies for reducing emissions and 3) the implications of exceeding a critical threshold of cumulated emissions on optimal emission policies in a stylized sequential emission problem.

In a nutshell, the results presented in section 5 support the conclusion that uncertainties about the implementability of decisions on emission reductions (or increases) call for more precautionary policies. By contrast, uncertainties about the implications of exceeding critical cumulated emission thresholds tend to make precautionary policies sub-optimal.

More specifically, the results of section 5 suggest that uncertainties about the implementability of decisions on emission reductions and, up to a more limited extent, uncertainties about the implications of exceeding critical cumulated emission thresholds have a grater impact on optimal emissions policies than uncertainties on the availability of effective technologies for reducing GHG emissions.

This is at the first glance perhaps a bit surprising but actually quite understandable: if decisions on emission (no matter whether reductions or increases) can be implemented with certainty, it is obviously better to delay necessary but costly reductions until available technologies make abatements cheaper. This holds as far as delays do not lead global emissions to exceed the critical threshold $crE$.

But if we cannot be sure that future decisions will be implemented with certainty – for instance, because of inertia in legislation or political instability – than starting implementing emission reductions (or trying doing so) sooner yields higher rewards. This is a typical case in which precautionary policies are optimal.

How earlier is it optimal to undertake costly abatement steps (rather than waiting for technological innovation to make emission reductions cheaper) very much depends on the rewards structure and on the uncertainties of the specific emission problem at stake.

Perhaps more surprisingly, the results of section 5.4 suggest that the optimal time for starting reducing emissions also depends on the level of uncertainty about the implications of trespassing critical thresholds of cumulated emissions. As these uncertainties increase, precautionary policies become sub-optimal. In other words: the better we can estimate the consequences of exceeding critical thresholds, the more does it pay off adopting precautionary policies.

Two caveats are in order here. First, while the results presented in section 5 are rigorous (the optimal emission policies our conclusions rely upon have been machine checked), the stylized emission problem for which we have computed such policies is defined in terms of a small but not empty set of parameters. In particular, the value of policy sequences (optimal or not) crucially depend on the problem rewards that is, on the values of the four parameters $badOverGood$, $highOverGood$, $lowOverGoodAvailable$ and $lowOverGoodUnavailable$, see Table 1 at the end of section 4. Are our conclusions only valid for these specific values?

Apart from substantiating our findings with a careful (but, necessarily, prohibitively expensive) sensitivity analysis, we can try to achieve a better analytical understanding of the role of the above parameters on optimal policy sequences.

From the definition of the reward function given at the end of section 4, we can immediately deduce that, at each decision step, the costs of selecting low emissions are greater or equal to

$$highOverGood - lowOverGoodAvailable$$

Remember that $lowOverGoodUnavailable$ is the ratio between the benefits of low emissions and the benefits of being in a good world when effective technologies for reducing GHG emissions are unavailable. Similarly, $lowOverGoodAvailable$ is the ratio between the benefits of low emissions and the benefits of being in a good world when effective technologies are available. As summarized in Table 1, we require $lowOverGoodUnavailable$ to be smaller or equal to $lowOverGoodAvailable$ (effective technologies for reducing GHG emissions diminish the costs of low emissions) and $lowOverGoodAvailable$ to be smaller or equal to $highOverGood$ (low emissions cost more than high emissions). Thus, the difference between $highOverGood$ and $lowOverGoodAvailable$ represents the minimal costs (e.g., due to missed growth, higher GHG filtering and sequestration costs, taxes, etc.) implied by low emission measures. By contrast, the costs (damages) that can be avoided by keeping the world in a good state are expressed, in our stylized decision problem by the difference

$$1 - badOverGood$$

Thus, if $1 - badOverGood$ is smaller or equal to $highOverGood - lowOverGoodAvailable$, selecting low emissions never pays off. Therefore,

$$crBadOverGood = 1 - (highOverGood - lowOverGoodAvailable)$$

is an important threshold in the parameters space of our emission problem: for values of $badOverGood$ between $crBadOverGood$ and one, selecting low emissions cannot be optimal: in this interval, optimal policies will recommend high GHG emissions. Are there other important thresholds in the problem's parameter space? At this point, we do not know. We have computed optimal policy sequences for a few values of $badOverGood$ between 0.8 and 0.91. These results confirm the analysis and support the conclusion presented above.

The second caveat is that the results presented in section 5.4 offer a rather limited view on the impacts of uncertainties about the implications of exceeding critical thresholds of cumulated GHG emissions on optimal policies. It is true that we have performed more assessments (with probabilities of 5% and 20% of turning to a bad world for cumulated emissions below $crE$, not reported section 5.4) and that these support the conclusions drawn above.

However, our statistics on the set of possible trajectories associated with a given policy sequence (optimal or not) has been throughout section 5 very rudimentary: we have only assessed the three most likely trajectories, their values and probabilities and the expected sum of rewards.

In studying the impacts of uncertainties about the implications of exceeding critical thresholds, we have to do with 51200 possible trajectories for every single policy sequence. In this case, a more comprehensive statistics would probably be at place. This is computationally challenging, see section 7.

Thus, the conclusions that we can draw from the results of section 5 are necessarily preliminary. Notice, however, that they are consistent with the analysis reported in Webster (2008) for a two-step decision problem. We are not aware of studies in

which the impact of uncertainties on optimal emission policies have been studied systematically for more than two decision steps.

It is probably fair to also point out that, as uncertainties on the implementability of emissions decisions increase and (therefore) optimal policies require more and more precautionary approaches, the advantages (in terms of rewards) of earlier emission reductions against delays do vanish: in the limit case in which political decisions have no bearing on the measures actually implemented, all policies are optimal.

It should also be remembered that, in our idealized problem, we have kept the cumulated emission threshold $crE$ and the critical number of decision steps for technological innovation $crN$ fixed. In increasing the uncertainty about the availability of technologies for reducing emissions and about the implications of exceeding $crE$, we have modified the probability distributions below and above $crN$ and $crE$ symmetrically. Thus, taking as reference the certain case, we have increased the probability that efficient technologies become available before $crN$ steps from zero to 0.1 and at the same time decreased the probability after $crN$ from one to 0.9. Similarly for uncertainties on the consequences of exceeding $crE$. It goes without saying that shifting $crN$ and $crE$ does indeed have a strong impact on optimal policies.

Thus, the results presented in section 5 do not imply that improving the accuracy of $crN$ and $crE$ estimates is not worth the efforts. But they suggests that obtaining more realistic estimates for the probability of effective technologies for reducing GHG becoming available before and after a critical date is perhaps not as crucial (for computing optimal emission policies for realistic decision problems) as improving our understanding of the implementability of decisions on emission reductions or increases.

Obtaining plausible estimates for the probabilities of being able to implement decisions on emissions reductions or increases naturally brings a political perspective into the problem of computing plausible optimal emission policies.

## 7  Future work

Realistic GHG emission problems involve more than one decision maker (countries) in a competitive situation rather than a single decision maker.

As explained in the introduction, a generic computational theory for SDPs under uncertainty, multiple players and a finite but unknown number of decision steps is, to the best of our knowledge, still missing. Developing such a theory is a challenging research program. The theory would have to span the border between control and game theory and likely require the introduction of new equilibrium notions. One promising approach towards developing a general theory of optimal decision making is to extend the formalization of SDPs presented in Botta et al. (2017b) using the notions of *quantifier* and of *selection function* (together with their respective products) introduced in Escardo and Oliva (2010); Hedges (2017) for infinite horizon open games.

From a more applicational point of view, there are two obvious ways in which the work presented in this paper could be extended to provide more useful insights into the problem of making optimal decisions on emission paths under uncertainty.

One would be to compute optimal emission policies for a realistic emission problem. Beside extending the notions of state and control spaces and, e.g., allow the decision maker to pick up a few intermediate emission levels between *Low* and *High*, this would require assessing the costs and the benefits of implementing a given emission level using a realistic integrated assessment model. Such an enterprise would require an interdisciplinary effort on the border between climate science and computing science. Technically, it would require extending the framework for the specification of SDPs

`SequentialDecisionProblems`[7] with a small domain specific language for emission problems.

Another way of extending the work presented in this paper would be to keep the focus on stylized emission problems like the one of section 4 but improve the statistical study of the logical consequences of taking decisions according to optimal policy sequences. This could yield to tools that support accountable decision making in real-time situations, for instance, during

negotiations. Technically, this would imply, among others, extending `SequentialDecisionProblems` with algorithms for computing all optimal policies for a given decision problem or perhaps just a certain number of optimal policies.

As we have seen at the end of section 5, computing optimal policies and parsing large collections of possible trajectories or "decision networks" can be computationally challenging even for idealized problems.

Thus, extending `SequentialDecisionProblems` for computing more optimal policy sequences and more compre-

hensive statistical analyses of decision networks would benefit from exploiting the concurrency inherent in many of the algorithms presented in Botta et al. (2017b). This is also an interdisciplinary enterprise involving formal methods (concurrent implementations should preserve the machine checkable optimality proofs that come with the sequential implementation), high-performance computing and climate science.

**Appendix A:  a summary of Botta et al. (2017a)**

The theory presented in Botta et al. (2017a) allows the specification of SDPs with uncertain outcomes and, for a specific problem, the computation of provably optimal policy sequences and of the possible consequences of taking decisions according to an arbitrary policy sequence.

As explained in the introduction, the theory is dependently typed and the formalization language is Idris, see Brady (2013). Here, we summarize the main requirements and the main results of the theory in a simplified form. For a more detailed

discussion of the notion of decision process, decision problem, monadic decision problem, uncertainty, reachability, viability, policy, policy sequence, possible trajectories and avoidability, we refer the reader to Botta et al. (2017a). In a nutshell, a monadic SDP can be specified in terms of the four functions already introduced in section 2:

$$State \quad : (t : \mathbb{N}) \rightarrow Type$$
$$Ctrl \quad : (t : \mathbb{N}) \rightarrow (x : State\ t) \rightarrow Type$$
$$next \quad : (t : \mathbb{N}) \rightarrow (x : State\ t) \rightarrow (y : Ctrl\ t\ x) \rightarrow M\ (State\ (t+1))$$
$$reward : (t : \mathbb{N}) \rightarrow (x : State\ t) \rightarrow (y : Ctrl\ t\ x) \rightarrow (x' : State\ (S\ t)) \rightarrow Val$$

---

[7] In https://gitlab.pik-potsdam.de/botta/IdrisLibs

$M$ here is a monad and represents the problem's uncertainties. For deterministic problems (no uncertainties), $M$ is equal to $Id$ and $next$ associates to a state-control pair a unique next state. For non-deterministic problems $M = List$ and for stochastic problems $M = Prob$. Since $M$ is a monad and therefore a functor, it is equipped with a function

$$fmap : (a \rightarrow b) \rightarrow M\ a \rightarrow M\ b$$

that maps functions of type $a \rightarrow b$ for arbitrary $a, b : Type$ to functions of type $M\ a \rightarrow M\ b$ and preserves identity and function composition. The type of the values returned by the reward function, $Val$, is required to be equipped with a zero value $zero : Val$, with an addition $(\oplus) : Val \rightarrow Val \rightarrow Val$ and with a total preorder $(\sqsubseteq)$. Moreover, $\oplus$ is required to be monotonic with respect to $(\sqsubseteq)$:

$$monotonePlusLTE : a \sqsubseteq b \rightarrow c \sqsubseteq d \rightarrow (a \oplus c) \sqsubseteq (b \oplus d)$$

As mentioned in section 2, a decision maker has also to specify a monotonous measure for weighting uncertain outcomes

$$meas \qquad : M\ Val \rightarrow Val$$
$$measMon : \{A : Type\} \rightarrow (f : A \rightarrow Val) \rightarrow (g : A \rightarrow Val) \rightarrow ((a : A) \rightarrow (f\ a) \sqsubseteq (g\ a)) \rightarrow (ma : Prob\ A) \rightarrow$$
$$meas\ (fmap\ f\ ma) \sqsubseteq meas\ (fmap\ g\ ma)$$

The functions $monotonePlusLTE$ and $measMon$ are examples of *specifications*: their types formulate properties that $\oplus$, $(\sqsubseteq)$ and $meas$ have to fulfill for the Botta et al. (2017a) theory to be applicable. In this appendix, we will see further examples of propositional types that encode notions, e.g., of optimality or, as in the case of $Bellman$, theorems of the Botta et al. (2017a) theory. With the notions of states and controls in place, one can formalize the notions of policy and policy sequence:

$$Policy : (t : \mathbb{N}) \rightarrow Type$$
$$Policy\ t = (x : State\ t) \rightarrow Ctrl\ t\ x$$

**data** $PolicySeq : (t : \mathbb{N}) \rightarrow (n : \mathbb{N}) \rightarrow Type$ **where**
    $Nil : PolicySeq\ t\ Z$
    $(::) : Policy\ t \rightarrow PolicySeq\ (S\ t)\ m \rightarrow PolicySeq\ t\ (S\ m)$

As discussed in section 2, policies are functions that associate controls to states. They are dependently typed because the domain of $Policy\ t$ depends on the decision step index $t$. Moreover, its codomain, $Ctrl\ t\ x$, depends on $t$ and on $x$.

Policy sequences are just sequences of policies. Since policies are dependently typed functions, we cannot simply collect them in a list or in a vector. The data declaration $PolicySeq$ completely defines the set of all possible policy sequences. In particular, a sequence can only be empty ($Nil$, for $n = 0$) or consist of a head (a policy for taking a decision step at an arbitrary decision step $t$) consed (in functional languages, the data constructor $(::)$ is called $Cons$, and $p :: ps$ is spelled $p$ "consed" with $ps$) together with a policy sequence for (for $n = S\ m = m + 1$) steps.

For a consistent theory of sequential decision making under uncertainty, the notions of policy and that of policy sequence actually have to be made more precise. This requires introducing the notions of reachability and viability. In this summary, we

omit these important but rather technical aspects, see Botta et al. (2017a). As explained in section 2, the notion of optimality for policy sequences is defined in terms of the measured sum of possible rewards. This is given by a value function

$$val : (x : State\ t) \rightarrow PolicySeq\ t\ n \rightarrow Val$$
$$val\ \{t\}\ \{n = Z\}\quad x\ ps\quad\quad = zero$$
$$val\ \{t\}\ \{n = S\ m\}\ x\ (p :: ps) = meas\ (fmap\ f\ mx')\ \mathbf{where}$$
$$\quad y\quad : Ctrl\ t\ x$$
$$\quad y\quad = p\ x$$
$$\quad f\quad : State\ (S\ t) \rightarrow Val$$
$$\quad f\ x' = reward\ t\ x\ y\ x' \oplus val\ x'\ ps$$
$$\quad mx'\ : M\ (State\ (S\ t))$$
$$\quad mx' = next\ t\ x\ y$$

Notice that when the policy sequence is not empty, the measure $meas$ has to be applied to the result obtained by adding $reward\ t\ x\ y\ x'$ (the reward obtained by selecting the control $y$ in $x$ and for the next state $x'$) to $val\ x'\ ps$ (the value of making $m$ decisions according to the policy sequence $ps$) for every $x'$ in $next\ t\ x\ y$. It is this recursive call of $val$ for every $x'$ in
$next\ t\ x\ y$ that makes the problem of evaluating policy sequences computationally intractable. For the case in which $State\ t$ is finite, one can recover linear complexity in $n$ via tabulation.

The value of policy sequences is the key for formalizing the notion of optimality for policy sequences: a policy sequence for $n$ decision steps is optimal iff no other sequence (also for $n$ decisions) attains a higher sum of possible rewards for any state:

$$OptPolicySeq : PolicySeq\ t\ n \rightarrow Type$$
$$OptPolicySeq\ \{t\}\ \{n\}\ ps = (ps' : PolicySeq\ t\ n) \rightarrow (x : State\ t) \rightarrow val\ x\ ps' \sqsubseteq val\ x\ ps$$

The main result of the theory presented in Botta et al. (2017a) is a verified, generic implementation of backwards induction:

$$bi : (t : \mathbb{N}) \rightarrow (n : \mathbb{N}) \rightarrow PolicySeq\ t\ n$$
$$bi\ t\ Z\quad\quad = Nil$$
$$bi\ t\ (S\ n) = optExt\ ps :: ps\ \mathbf{where}$$
$$\quad ps : PolicySeq\ (S\ t)\ n$$
$$\quad ps = bi\ (S\ t)\ n$$
$$biLemma : (t : \mathbb{N}) \rightarrow (n : \mathbb{N}) \rightarrow OptPolicySeq\ (bi\ t\ n)$$

The implementation of $biLemma$ relies on the notion of optimal extension of a policy sequence

$$OptExt : PolicySeq\ (S\ t)\ m \rightarrow Policy\ t \rightarrow Type$$
$$OptExt\ \{t\}\ \{m\}\ ps\ p = (x : State\ t) \rightarrow (p' : Policy\ t) \rightarrow val\ x\ (p' :: ps) \sqsubseteq val\ x\ (p :: ps)$$

and on a formal proof (that is, a total implementation) of Bellman's principle of optimality (Bellman, 1957):

$$Bellman : (ps : PolicySeq\ (S\ t)\ m) \rightarrow OptPolicySeq\ ps \rightarrow (p : Policy\ t\ (S\ m)) \rightarrow OptExt\ ps\ p \rightarrow OptPolicySeq\ (p :: ps)$$

As usual when encoding propositions through types we read *Bellman* as the first-order logic proposition: for every policy sequence *ps* and every policy *p*, if *ps* is optimal and *p* is an optimal extension of *ps*, then *p*::*ps* is optimal. The implementation of *bi* relies on *optExt*: this is a function that takes a policy sequence and computes one of its optimal extensions:

$$optExt \quad\quad : PolicySeq\ (S\ t)\ n\ \rightarrow\ Policy\ t\ (S\ n)$$
$$optExtLemma : (ps\ :\ PolicySeq\ (S\ t)\ n)\ \rightarrow\ OptExt\ ps\ (optExt\ ps)$$

Thus, computing the optimal extension of a policy sequence of type *PolicySeq* (*S t*) *n* implies solving an optimization problem for every state in *State t*. If the set of control *Ctrl t x* is finite for a given $x\ :\ State\ t$, this problem can be solved by linear search. A further result of the theory presented in Botta et al. (2017a) is a generic algorithm for computing all possible trajectories that can be obtained by applying a policy sequence (optimal or not) starting from a given state or, if the decision
maker takes decisions under imperfect information, from an *M*-structure of states. In both cases, the result is an *M*-structure of state-control sequences:

$$\textbf{data}\ StateCtrlSeq\ : (t\ :\ \mathbb{N})\ \rightarrow\ (n\ :\ \mathbb{N})\ \rightarrow\ Type\ \textbf{where}$$
$$Nil\ :\ (x\ :\ State\ t)\ \rightarrow\ StateCtrlSeq\ t\ Z$$
$$(::)\ :\ \Sigma\ (State\ t)\ (Ctrl\ t)\ \rightarrow\ StateCtrlSeq\ (S\ t)\ n\ \rightarrow\ StateCtrlSeq\ t\ (S\ n)$$
$$possibleStateCtrlSeqs \quad\quad : (x\ :\ State\ t)\ \rightarrow\ (ps\ :\ PolicySeq\ t\ n)\ \rightarrow\ M\ (StateCtrlSeq\ t\ n)$$
$$morePossibleStateCtrlSeqs : (mx\ :\ M\ (State\ t))\ \rightarrow\ (ps\ :\ PolicySeq\ t\ n)\ \rightarrow\ M\ (StateCtrlSeq\ t\ n)$$

For the implementations of *biLemma*, *Bellman*, *possibleStateCtrlSeqs* and *morePossibleStateCtrlSeqs* we refer the reader to Botta et al. (2017a). In section 5 we make extensive usage of, among others, *bi* and of *possibleStateCtrlSeqs* for computing optimal emission policies and possible state-control sequences.

*Acknowledgements.* The authors thank the ESD editors and reviewers, whose comments have lead to significant improvements of the original manuscript. The work presented in this paper heavily relies on free software, among others on Idris, Agda, GHC, git, vi, Emacs, LATEX and on the FreeBSD and Debian GNU/Linux operating systems. It is our pleasure to thank all developers of these excellent products. This work was partially supported by the projects GRACeFUL (grant agreement No 640954) and CoeGSS (grant agreement No 676547), which have received funding from the European Union's Horizon 2020 research and innovation programme.

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
