# Peer review of "The impact of uncertainty on optimal emission policies"

_Earth System Dynamics, 2017_

## Author Comment (AC1) · 13 Oct 2017

The results presented in our contribution naturally raise an important question and a legitimate methodological criticism.

We have not expanded on this question (and on the related criticism) in the manuscript, we thought that it would be a good subject for an interactive discussion.

In our contribution, we have argued that uncertainties about the implementability of decisions on emission reductions (or increases) call for more precautionary policies but uncertainties about the implications of exceeding critical cumulated emission thresholds make precautionary policies sub-optimal.

If recognized, such results have obvious consequences both for emission policy mak-

ing and for scientific research planning: if one accepts that turning political decisions into effective legislations is intrinsically uncertain, our study provides a very strong argument in support for precautionary policies and early (earlier) emission reductions.

Similarly, if one wants to justify precautionary policies under uncertainties about the implications of exceeding critical cumulated emission thresholds, one has to try to reduce such uncertainties, e.g. through research programs specifically aimed at sharpening our understanding of potentially dangerous climate thresholds.

Our results are rigorous in the sense that the optimal emission policies our conclusions rely upon are provably optimal. The stylized emission problem for which we have computed such policies, however, is defined in terms of a small but not empty set of parameters. In particular, the value of policy sequences (optimal or not) crucially depend on the problem rewards that is, on

- $badOverGood$

- $lowOverGoodUnavailable$

- $lowOverGoodAvailable$

- $highOverGood$

The optimal policy sequences discussed in section 5 have been computed for the specific (albeit stylized) emission problem in which these parameters take the values 0.5, 0.1, 0.2 and 0.3, respectively.

Are our conclusions only valid for these specific values?

Rather than trying to substantiate our findings with a (necessarily rudimentary) sensitivity analysis, we can try to achieve a better analytical understanding of the role of the above parameters on optimal policy sequences.

We have defined the reward function of our stylized problem at page 15 of Section 4 "A stylized sequential emission problem". From the definition we can immediately deduce that, at each decision step, the costs of selecting low emissions are greater or equal to

$$highOverGood - lowOverGoodAvailable$$

Remember that $lowOverGoodUnavailable$ is the ratio between the benefits of low emissions and the benefits of being in a good world when effective technologies for reducing GHG emissions are unavailable.

Similarly, $lowOverGoodAvailable$ is the ratio between the benefits of low emissions and the benefits of being in a good world when effective technologies are available.

We obviously require $lowOverGoodUnavailable$ to be smaller or equal to $lowOverGoodAvailable$ (effective technologies for reducing GHG emissions diminish the costs of low emissions) and $lowOverGoodAvailable$ to be smaller or equal to $highOverGood$ (low emissions cost more than high emissions).

Thus, the difference between $highOverGood$ and $lowOverGoodAvailable$ represents the minimal costs (e.g., due to missed growth, higher GHG filtering and sequestration costs, taxes, etc.) implied by low emission measures.

By contrast, the costs (damages from climate change) that can be avoided by keeping the world in a good state are expressed, in our stylized decision problem by the difference

$$1 - badOverGood$$

Thus, if $1 - badOverGood$ is smaller or equal to $highOverGood - lowOverGoodAvailable$, selecting low emissions never pays off. Therefore,

$$crBadOverGood = 1 - (highOverGood - lowOverGoodAvailable)$$

is an important threshold in the parameters space of our emission problem: for values of $badOverGood$ between $crBadOverGood$ and one, selecting low emissions cannot be optimal: in this interval, optimal policies will recommend high GHG emissions.

Are there other important thresholds in the problem's parameter space? We do not know. As mentioned above, the results discussed in section 5 have been computed with $badOverGood$ equal to 0.5 and $crBadOverGood$ equal to 0.9.

Additionally, we have computed optimal policy sequences for some values of $badOverGood$ between 0.5 and 0.91. These results confirm the analysis presented above and support the conclusion that uncertainties about the implementability of decisions on emission reductions call for more precautionary policies but uncertainties about the implications of exceeding critical cumulated emission thresholds tend to make precautionary policies sub-optimal.

Perhaps you have a set of parameters values for which you would like to see how optimal policies look like for the cases discussed in section 5?

In this case, please let us know. We will compute the policies and post the results.

---

## Referee Comment (RC1) · Anonymous Referee #1 · 21 Nov 2017

**General comments**

I find the overall approach taken in Botta et al. (2017a,b) very innovative and I am happy to see an application thereof. The current paper considers an interesting and important question. In the following, I provide several suggestions that might improve/clarify the paper.

0. In the context of optimal emission policies, I wonder whether the model of a single decision-maker being concerned with the dynamic implementation of policies is the most relevant setting. It seems that the overall coordination problem could be much more important. There are of course applications, where the single-decision maker being concerned with the dynamic implementation and its consequences is critical: For instance, nuclear power clearly has a key time dimension for the sequence of moves;

[Figure]

so have other economic policies such as competition policy. In both cases, arguably a single decision-maker has a lot of influence on the overall policy. It is not clear to me whether this is the case for global emission policies and it should be argued more forcefully by the authors why it is relevant.

1. More importantly though, one can also turn the question in 0 around and ask: Is this actually the adequate framework to consider the question at hand? It is in principle possible to consider dynamic game theoretic models to discuss the timing of actions *and* their strategic interrelation. There can be good reasons not to follow this path here but it should be made clear why.

2. To me a distinguishing feature of the current approach is its focus on correct implementation. I consider this aspect to be of great importance and often neglected in policy advice (and in academic work as well). Why is this important for the particular problem at hand?

3. Overall, I am left wondering what the exact contributions to the literature is. I am fully aware that, working at the intersection of different fields, making this clear to researchers from different backgrounds is hard. I may miss the obvious - not being familiar with certain aspects of emission policy research - but I would still urge the authors to be more concrete about this.

4. I find the scattered comments on the different aspects of the papers Botta et al. (2017a,b) rather confusing. An alternative could be to have a section where the necessary information from the previous papers is summarized; then followed by the application of this paper here. It may also help to have a technical appendix where the key ideas are summarized.

**Specific comments**

0. The introduction states as a key result that different kinds of uncertainty can be distinguished. This is vague. How do these different kinds of uncertainty differ precisely?

Do they refer to the different nature of uncertainty, i.e. a distinction of probabilistic versus non-deterministic uncertainty? And in which sense can this framework "seamlessly" deal with it (see also point 2 below)?

1. Section 2.2. describes the reward function. I find this discussion hard to follow; again a more systematic introduction of the tools of the previous papers would be helpful here. An alternative could also be to describe the decision problem as a sequence of payoffs, which the decision-maker seeks to maximize, and then turn to the control theoretical perspective of viewing this as a step-by-step process.

2. Section 5 analyses the effect of different types of uncertainty. This looks rather ad-hoc to me as it is unclear in which sense the derived conclusions ("implementability risk is more important") are driven by particular parameter choices. What is the argument that it is a robust result? What about a more systematic approach including a sensitivity analysis?

**Technical corrections**

Typos:

p. 10, l. 6 "the the"

p. 10, l. 25 "Ionescu Inonescu"

p. 12, l. 24 "Without losT of generality"

p. 14, l. 24 "in termS of"

p. 20, l. 29 "explained:the

---

## Editor Comment (EC1) · M. Crucifix (Editor) · 8 Dec 2017

Based on the review and my own reading I can see that the article as a welcome application of formal decision theory to the problem of optimal decision policies. Given that formal decision theory is not so well know in climate and earth system sciences, the article has partly a pedagogic function and it fulfills this role.

One possible hurdle (for many readers I suspect) is the functional language notation. It was not entirely clear to me what was standard functional language notation (if there is such a standard), and what was more idiosyncratic to the Idris language. The first serious hurdle is the expression of the monotonicity condition p.10. For example, the functional mapping `fmap` will be unknown to most readers, and in fact the whole ex-

pression will be challenging. It would be a shame to have readers abandoning reading at this point. I can see two (non-exclusive) solutions: spend more time in explaining the notation, the context and the fact that the article comes with a `git` repository with Idris code, and (2) put the formal expression of the monotonicity condition in an appendix, with due explanations. The authors may also point where, in the git code, this condition is expressed (file name and line number).

The authors do a good job in explaining the interest of formal decision theoretic approaches. As they correctly point out, the choice of a measure function mapping probabilities to a reward is subjective and important. There is a vast literature on this particular point which cannot be entirely reviewed here, but it would be useful to have a few more words making explicit the rationale for using the minimum, the maximum, or the expectation.

The fact that high emissions are always optimal at the last time step is interesting. It comes from the fact that there will not be a following time step: this is the end of the time line and future generations are infinitely discounted. It sounds like the proverbial prodigality of the one who just learned she is ruined (in French: "foutu pour foutu"). Perhaps this is a nice example (which deserves an observation) of the fact that convenience assumptions (like the finite number of decision steps) generate outcomes that ought to be rightly interpreted as consequences of these convenience assumptions. This is a surely trivial or extreme example of the more general fact that the output of a decision-support algorithm should not be delivered without further guidance to the decision-maker.

In a revised version, the authors may also want to introduce some of their own comments (response to a legitimate criticism), along with addressing the concerns of reviewer 1. The article is then likely to be acceptable for publication.

See some editorial comments in the attached document. There are a few more typos which will be corrected at the copy-editing stage.

Please also note the supplement to this comment:
https://www.earth-syst-dynam-discuss.net/esd-2017-86/esd-2017-86-EC1-
supplement.pdf

**Supplement:**

[revised manuscript text omitted]

---

## Referee Comment (RC2) · M. Crucifix (Referee) · 22 Dec 2017

Comments on this paper have been expressed in the editorial comment.

———————————————

---

## Author Comment (AC6) · 27 Dec 2017

Please, cf. response to the editorial comment EC1.

———————————————

---

## Author Response (AR1)

**Author's response**

We have resubmitted a revised version of the original manuscript.

Point-by-point replies to the comments RC1 (21 Nov. 2017), EC1 (08 Dec. 2017) and RC2 (22 Dec. 2017) have been already provided in AC3 (06 Dec. 2017), AC5 (14 Dec. 2017) and AC6 (27 Dec. 2017), respectively, see discussion. For completeness, we have included these replies in the "Point-by-point replies to . . . " sections below.

The revised version accounts for the recommendations and for the corrections pointed out by the reviewers. In particular, we have implemented the obligations 01 to 10 from our AC5 "Answers to the editorial review" from 14 Dec. 2017. The major modifications w.r.t. the original submission are reported in section "Major revisions".

We have uploaded a marked-up diff between the original and the revised manuscript as a supplement to our submission, see ESD-2017.submission.diff.R01.R02.pdf.

**Point-by-point replies to RC1, see also AC3 on discussion.**

Thank you for the detailed review and suggestions. In the following, we provide point-to-point answers to the general comments 0 to 4 and to the specific comments 0 to 2.

**Generic comments 0 and 1:**

We have failed to make the point clear in the introduction: one would of course like to tackle the problem of computing optimal emission policies for individual countries as a (mixed sequential and simultaneous) sequential coordination game with a finite number of decision makers over a finite (but not necessarily known) number of decision steps and under different sources of uncertainty.

To the best of our knowledge, no theory (let apart a computational theory) is available for such problems today. A very common approach is that of slicing the problem into two main questions:

   a) *When* and by *how much* global GHG emissions should be reduced to avoid unmanageable future states?

   b) *How* to make sure that (fair, optimal, etc.) emission reduction quotas consistent with given optimal global reduction are actually implemented by individual countries or regions?

Answers to a) can be sought, among others, by extending standard control-theory approaches (one decision maker) to sequential decision problems with uncertain (non-deterministic, stochastic, fuzzy) outcomes.

Answers to b) can be sought, among others, by extending standard game-theory approaches (multiple decision makers) to decision games under mechanisms for incentivating the emergence of trust, coalitions and binding agreements.

From this perspective, "solving the GHG emission problem" requires an iterative solution of a) and b). Again, to the best of our knowledge, no attempts have been done so far at coupling a) and b) and solving the full problem. In our contribution, we focus the attention on a).

In revising our manuscript, we will expand the introduction and make the context of our contribution more clear.

**Generic comment 2:**

Using a verified computational method for computing optimal policies is crucial simply because optimality (e.g., of supposedly optimal policies) cannot, in general, be tested. This is one of the most prominent examples where "proving" is easier than "testing". From an applicational perspective, computing verified policies allows us to study the impact of different assumptions (e.g. about uncertainties) in a rigorous fashion. In revising our manuscript, we will discuss this point in more detail in section 3.

**Generic comment 3:**

In revising our manuscript, we will make the context of our contribution more clear and compare our results to, among others, those presented in the works of works of M. Webster.

**Generic comment 4:**

We are going to summarize the results of Botta et al. (2017a,b) in an appendix of our revised manuscript.

**Specific comment 0:**

The theory presented in Botta et al. (2017a,b) is based on the notion of *monadic* dynamical systems originally introduced by Ionescu in his PhD thesis. This allows us to treat deterministic, non-deterministic, stochastic, fuzzy, etc. problems with a seamless approach: the differences are captured by a single problem parameter and all computations (e.g. of optimal policies, possible trajectories, rewards, etc.) are generic with respect to this parameter. In revising our manuscript, we will make this point more clear.

**Specific comment 1:**

A sequential decision problem cannot be described as a sequence of payoffs: one has to give a function that returns one payoff for every suitable combination of current state, selected control and next possible state. We will summarize the results of Botta et al. (2017a,b) in an appendix of our revised manuscript and make this point more clear.

**Specific comment 2:**

This is a very important criticism that we have tried to anticipate with a "On a legitimate criticism to our contribution" comment posted on Oct. 13 on the ESD discussion site. Is there something specific that you find unconvincing in our comment? If the comment helps dissipating some of your concerns about the robustness of the results presented in section 5, we would be pleased to add a revised version of the comment to section 5 of our revised manuscript.

**Point-by-point replies to EC1, see also AC5 on discussion.**

Thank you for the review and for the detailed comments and corrections of the supplementary document!

We are going to prepare a major revision of the original manuscript and implement your recommendations and those of Referee 1. In the following, we have listed a number of TODOs. The idea is to provide you with an account of our revision plans. We will use the list as a guideline for revising our original manuscript. If new reviews and comments will become available, we will update the list accordingly.

TODO (first manuscript revision, status 2017.12.13):

0. Correct typos and errors according to RC1 and EC1 (supplement).

1. Explain more clearly the differences between plain mathematical notation (e.g., set comprehension in $T = A, U$), functional notation (e.g., $State : Nat-> Set$) and Idris specific formulas (EC1). Perhaps introduce a short "Notation" section after the introduction and before section 2 "Sequential emission problems"? Explain that the article comes with a git repository and give the URL of IdrisLibs (EC1).

2. Summarize the results of Botta et al. (2017a,b) in an appendix of the revised manuscript (RC1.GC4). Move the formal monotonicity condition to the appendix. There, give the type of fmap and an example, e.g., for lists (EC1).

3. Discuss the differences between *best*, *worst* and *average* (expected value) as measures of uncertainty in more detail (EC1). Perhaps link this discussion to the problem of finding sensible influence (responsibility) measures in sequential decision problems under uncertainty.

4. When discussing basic facts about optimal emission policies (in the beginning of section 5), stress the importance of making decision makers aware of the consequences of (often implicit) assumptions. In particular, explain that the last decision step needs a special care if one wants to avoid apparently inconsistent results (reducing emission at the last step is never optimal, EC1) or just account for meaningful boundary conditions. Perhaps formulate a sustainability principle?

5. In the introduction, explain more clearly that one would like to tackle the problem of computing optimal emission policies for individual countries as a (mixed sequential and simultaneous) coordination game with a finite number of decision makers over a finite (but not necessarily known) number of decision steps and under different sources of uncertainty (RC1.GC1). Recall that (to the best of our knowledge), no theory (let apart a computational theory) is available for such problems and that a very common approach is that of slicing the problem into the questions:

   a) *When* and by *how much* should global GHG emissions be reduced to avoid unmanageable future states?

   b) *How* to make sure that (fair, optimal, etc.) emission reduction quotas, consistent with given optimal global reduction, are actually implemented by individual countries or regions?

   which, in a holistic approach, would have to be answered simultaneously. Answers the role of control-theory and of game-theory in a), b).

6. In section 3, explain in more detail that applying a verified computational method for computing optimal policies is crucial because optimality (e.g., of supposedly optimal policies) cannot, in general, be tested (sometimes proving is easier than testing, RC1.GC2).

7. Make the context of our contribution more clear and compare our results to, among others, those presented by M. Webster (RC1.GC3).

8. Explain (when referring to the new appendix, see TODO 02.) that the theory presented in Botta et al. (2017a,b) is based on the notion of *monadic* dynamical systems originally introduced by Ionescu in his PhD thesis. Explain that monads allows one to treat deterministic, non-deterministic, stochastic, fuzzy, etc. problems with a seamless approach in which the differences are captured by a single problem parameter and all computations (e.g. of optimal policies, possible trajectories, rewards, etc.) are generic with respect to this parameter (RC1.SC0).

9. In summarizing the results of Botta et al. (2017a,b) in an appendix (see TODO 02.), explain that a sequential decision problem cannot be described as a sequence of payoffs: one has to give a function that returns one payoff for every suitable combination of current state, selected control and next possible state (RC1.SC1).

10. Add a revised version of AC1 (comment "On a legitimate criticism to our contribution") to section 5 (6?) of the revised manuscript (RC1.SC2).

**Point-by-point replies to RC2, see also AC6 on discussion.**

Please, cf. response to the editorial comment EC1.

**Major revisions**

- Added appendix with a summary of the Botta2017,JFP theory [TODO 02, 09].

- Added a sentence to the introduction arguing that, to the best of our knowledge, no theory is currently available for tackling the problem of computing optimal emission policies for individual countries as a mixed sequential and simultaneous coordination game with a finite number of decision makers, over a finite but not necessarily known number of decision steps and under different sources of uncertainty [TODO 05].

- Added a "Notation" subsection to the introduction. Moved remarks on currying and dependent types from section 2 to "Notation". [TODO 01] Explained why we use dependently typed formalisations and Idris [Recommendation from the supplement to the editorial review EC1].

- Added a short discussion on alternative uncertainty measures (*worst*, *best*, etc.) after the introduction of the measure function [TODO 03].

- Expanded the discussion on measures that violate the monotonicity condition [Recommendation from the supplement to the editorial review EC1]

- Added remark at the beginning of section 5 to clarify the statement that "no optimal policy sequence can require selecting low emissions when the state of the world is bad" [Recommendation from the supplement to the editorial review EC1] and to comply with TODO 03.

- Coloured controls in state-control trajectories to make the impact of different policies more visible.

- Expanded the first part of the "Conclusions" section to integrate the comments of "On a legitimate criticism to our contribution" [TODO 10].

- Added two sentences to the "Conclusions" section to compare our results with those obtained by M. Webster [TODO 07].

- Rephrased the beginning of section "Future work" to remind the reader that, to the best of our knowledge, a generic computational theory for SDPs under uncertainty, multiple players and a finite but unknown number of decision steps is still missing [TODO 05].

- Added a sentence to section "Logical consequences" to explain in more detail that applying a verified computational method for computing optimal policies is crucial because optimality cannot be tested [TODO 06].

[revised manuscript text omitted]

---

## Author Response (AR2)

**Author's response**

We have addressed the technical corrections/clarifications listed in the Editor's decision from 17 March 2018 and submitted a revised version of the manuscript.

In the following we provide a point-by-point response to the Editor's comments and list the relevant changes introduced with our revision.

**p. 3, l.26 : p. 2 l. 21: use "\citep" (bracket references) where needed. Check manuscript throughout**

Done.

**p. 11, l.29 : "But notice that, as discovered by Ionescu (2009) 30 in the context of vulnerability studies, measures that pick up the most (least) probable Val-value of a probability distribution do violate the monotonicity condition" -> If the Ionescu thesis available online? If not, you might need to give a bit more detail. I tried to find a counterexample (of most-probable Val violating monotonicity) and, at least in a few minutes, I couldn't find any. Is there any simple counterexample you could give?**

The thesis is available online and we have provided the URL from which it can be downloaded in the references of our revision. A detailed explanation of why most/least probable measures violate monotonicity is given at pages 112-116 of Ionescu's thesis for the more general case in which meas is of type M X -> Y. We have referred to these pages in the revision of the manuscript. A counterexample for the case in which meas is of type M Val -> Val with Val representing natural numbers can be built as follows. Consider the probability distribution

ma = [(x, 0.3), (y, 0.3), (z, 0.4)]

with A = {x,y,z} and 0.3, 0.3 and 0.4 the probabilities of x, y and z, respectively. Let Val = Nat and f, g : A -> Val with

f x = 1   f y = 2   f z = 3

g x = 2   g y = 2   g z = 3

We have that $f\ a \leq g\ a$ for all a : A and

map f ma = [(1, 0.3), (2, 0.3), (3, 0.4)]

map g ma = [(2, 0.3), (2, 0.3), (3, 0.4)]

The most probable Val value of map f ma is 3 with probability 0.4. The most probable Val value of map g ma is 2 with probability 0.3 + 0.3 = 0.6. Thus, if we take meas to be the most probable value we have

meas (map f ma) = 3

meas (map g ma) = 2

in contradiction with measMon at page 27 of the revised manuscript. A similar counterexample can be constructed for the case in which meas takes the least probable event. Thus, most/least probable measures violate the monotonicity condition from page 27: it is possible to increase all elements of a probability distribution (possibly by zero increments) and having their measure to strictly decrease!

The counterexample is interesting for applications (taking the most probable event seems, at the first glance, a quite natural way of measuring uncertainties) but it does not add anything substantial to the study of the impacts of uncertainties on optimal emission policies. Thus, we have decided not to add it to our revision.

**p. 4., l, 21: programmes machine checked to be correct. This seems to be a central notion, but the average reader will need to undrestand better in what sense the programme is checked to be 'correct'. What does 'correct' mean in this context?**

We agree that an explanation is needed and have added a whole paragraph with a simple example (that of a program that computes the square root of a number) at page 4, lines 23-31 and at page 5 lines 1-10 of the revised manuscript.

**p. 4, l 18 : specificy the meaning of the $\sqsubseteq$ symbol. I do not understand why in this case we can't simply use $\leq$ (see also p. 26 l. 13). Is this is a generalisaton of $\leq$ for arbitrary defined order relations?**

No, there is no generalization but we have to distinguish between $(\leq)$ : Nat -> Nat -> Bool and $(\sqsubseteq)$ : Nat -> Nat -> Type: on the right hand side of BoundedBy n ms, All expects a value of type Nat -> Type. Here $(\sqsubseteq)$ is just a short cut (that we hope readers will relate to $(\leq)$) for the standard Idris data type LTE. This is defined as follows:

data LTE : Nat -> Nat -> Type where

LTEZero : LTE Z right

LTESucc : LTE left right -> LTE (S left) (S right)

The data type LTE defines what it means for a natural number m to be smaller than another natural number n. This is in contrast with $(\leq)$ : Nat -> Nat -> Bool that represents a computation that is, a decision procedure for LTE. The idea of LTE is that it defines a proof that m is smaller than n as a value that can be constructed in one of two disjoint ways. These correspond to the constructors LTEZero and LTESucc:

1) If m is zero it is just LTEZero with no arguments. This formalizes the idea that zero is smaller or equal to any natural number.

2) If m is the successor of a number m' and n is the successor of a number n' then a proof that m is smaller than n can only be constructed (by applying LTESucc) if one already has a proof that m' is smaller than n'.

The definition of LTE is inductive in much the same way as that of natural numbers in functional languages:

data Nat : Type where

Zero : Nat

Succ : Nat -> Nat

To make a concrete example: one can implement a proof that 2 is smaller than 42

prf : LTE 2 42

by applying LTESucc to a proof that 1 is smaller than 41

prf = LTESucc prf' where

prf' : LTE 1 41

prf' = LTESucc LTEZero

Here, we have applied LTESucc to a proof prf' that 1 is smaller than 41. In turn, prf' is constructed by applying once more LTESucc to a proof that 0 is smaller than 40. The latter is simply LTEZero.

We think that it would be too confusing to discuss these technicalities in the manuscript and the discussion would not bring any better understanding of the impacts of uncertainties on emission policies,

Such a technical level would probably also not be fair: at page 3 line 30, we have written that we do not assume our readership to be familiar with dependent types. Indeed, most of the work that we have done in preparing our original contribution was geared towards hiding the technicalities of the theory behind the application that we present.

Thus, in the revised manuscript, we have expanded a little bit on $(\sqsubseteq)$ at lines 20-13 of page 4 but we have not introduced further technical details.